# Analyzing Multilingualism in Large Language Models with Sparse Autoencoders

**Ikhyun Cho & Julia Hockenmaier**
Siebel School of Computing and Data Science
University of Illinois at Urbana-Champaign
Illinois, IL 61801, USA
{ihcho2, juliahmr}@illinois.edu

## Abstract

Despite the impressive multilingual capabilities of recent large language models (LLMs), the mechanisms underlying their language-specific processing remain largely unclear. In this paper, we investigate how LLMs handle multilingualism through the lens of sparse autoencoders (SAEs), uncovering distinctive patterns that offer new insights into their internal workings. Specifically, we introduce two novel concepts—*task instruction–focused (TF)* and *heading-focused (HF)* SAE features—and use them to reveal intrinsic discrepancies between high- and low-performing languages. Our analysis yields several key findings: (1) SAEs provide concrete evidence that LLMs have a precise understanding of prompt structure; (2) heading keywords (e.g., "Question," "Choices," and "Answer") play a distinct role in LLM processing; and (3) low-performing languages exhibit a relative deficiency in TF features compared to high-performing languages.

Building on these insights, we propose two practical strategies to improve zero-shot multilingual performance: (1) incorporating English heading keywords and (2) amplifying TF features through steering. Our approach improves zero-shot performance in low-performing languages by up to 3.7% on average on ARC-Challenge and MMLU, while also shedding new light on fundamental differences between high- and low-performing languages in LLMs. Our code is available at https://github.com/ihcho2/SAE-ML.

## 1 Introduction

Large language models (LLMs) have revolutionized the field of natural language processing, demonstrating remarkable performance across a wide range of tasks. Among these, their multilingual capabilities have attracted significant attention. Extensive research has been dedicated to enhance multilingual performance, such as language-specific finetuning (Adelani et al., 2021; Wilie et al., 2020), injecting language capabilities through continual learning (Shi et al., 2024; Cahyawijaya et al., 2023), adapter-based fine-tuning (Yong et al., 2022), prompting (Li et al., 2023b), among others. However, despite their impressive performance, the precise underlying mechanisms that enable LLMs to handle multilingual tasks remain elusive. Gaining deeper insights into these processes is crucial, not only for advancing our theoretical understanding of LLMs but also for addressing the persistent performance gaps across different languages (Cahyawijaya et al., 2024).

A promising direction in this space is cross-lingual in-context learning (X-ICL) (Winata et al., 2021), which aims to enhance target language performance by transferring knowledge from a pivot language, leveraging the robust in-context learning capabilities of LLMs (Brown et al., 2020). This approach is computationally efficient—requiring no parameter tuning—and provides a direct means to explore various aspects of multilingualism in LLMs. In this study, we extend this line of research by incorporating sparse autoencoders (SAEs), an emerging tool in mechanistic interpretability, to further investigate the multilingual processing mechanisms of LLMs.

SAEs, an entirely unsupervised approach, are rapidly emerging as a key tool in mechanistic interpretability. By using a sparsity loss term to reconstruct input embeddings (i.e., LLM dense embeddings), SAEs have proven effective—albeit to some extent—in decomposing these embeddings into sparse features, each corresponding to human-understandable concepts (Sharkey & Beren, 2022; Bricken et al., 2023). In this work, we demonstrate that this effective tool can yield novel insights into the underlying mechanisms of multilingualism in LLMs. Specifically, we analyze SAE patterns using parallel corpora, multilingual-ARC (Lai et al., 2023) and multilingual-MMLU (OpenAI, 2024), spanning multiple languages, seeking for distinct, consistent patterns across languages.

Our analysis reveals several key findings:

1. SAEs provide concrete and tangible evidence that LLMs have a precise grasp of prompt structure, effectively distinguishing between headings, task instructions, and test example segments. Our ablation results show that this understanding is impressively accurate.

2. We introduce two types of SAE features—task-instruction-focused features (TF features) and heading-focused features (HF features)—and identify fundamental discrepancies in these features between high- and low-performing languages, offering novel insights into their intrinsic differences.

3. Specifically, low-performing languages exhibit a relative lack of TF features compared to high-performing languages. Notably, we observe a strong correlation between the number of TF features and overall performance, highlighting their critical role in language-specific performance.

4. SAEs reveal that headings play a distinct role in LLM processing, as evidenced by the presence of multiple features that primarily activate on headings (i.e., HF features).

Based on above findings, we present two practical methods to enhance multilingual performance:

1. **English Heading Prompts:** Motivated by the distinctive role of headings in LLMs, we demonstrate that using English headings consistently enhances performance in low-performing languages. This approach is based on the rationale that, since LLMs are predominantly English-centric and treat English as a pivot language, they can more effectively leverage the critical function of headings when presented in English rather than in low-performing languages.

2. **Amplifying TF Features:** We show that amplifying the identified English TF features can enhance performance across both high- and low-performing languages. Notably, this approach not only improves low-performing languages but also high-performing languages, verifying the crucial role of TF features in LLMs' overall language performance.

By combining both approaches, we achieve average performance gains of up to 3.70% on m-MMLU and 1.94% on m-ARC for low-performing languages.

In summary, our main contribution is the introduction of two new types of SAE features (TF and HF features) which offers fresh insights into the fundamental discrepancies across languages in LLMs.

## 2 Related Work

### 2.1 Approaches to Understanding Multilingualism in Large Language Models

Various approaches have been proposed to analyze multilingualism in LLMs. For instance, some studies have explored multilingual reasoning abilities through self-attention layers (Hou et al., 2023; Li et al., 2023a), while others have identified language-specific neurons (Zhao et al., 2024) or examined language-centricity using the decoder-lens methodology

(Kargaran et al., 2024; Geva et al., 2022). Another promising and widely adopted approach involves leveraging the robust in-context learning (ICL) capabilities of LLMs (Winata et al., 2021).

In-context learning (ICL)—the emerging ability of LLMs to tackle a variety of tasks by leveraging just a few exemplars within a prompt—has become a dominant paradigm in natural language processing (Brown et al., 2020). Building on this capability, cross-lingual ICL (X-ICL) (Winata et al., 2021) aims to enhance performance in a target language by transferring knowledge from a pivot language. Recent studies have explored various strategies to improve target language performance, which is often low-resource. For example, Tanwar et al. (2023) show that semantically similar exemplars are crucial for effective language transfer, while Winata et al. (2022) find that mixing exemplars from multiple languages can also enhance performance. Cahyawijaya et al. (2024) propose in-context query alignment, which selects exemplars most similar to the test query and incorporates their translation pairs to further improve ICL performance in low-resource languages. Zhang et al. (2024) argue that the effectiveness of language transfer largely depends on the quality of the prompt template. Additionally, Tu et al. (2025) demonstrate that simply exposing models to multiple languages could sometimes improve performance regardless of the precise context.

While these works predominantly focus on the X-ICL framework—a cross-lingual transfer setup in which knowledge is transferred from a pivot language to a target language via in-context learning—we shift our focus to the zero-shot setting. Our primary objective is not to facilitate transfer, but rather to investigate the fundamental discrepancies between languages. We believe the zero-shot setting is better suited for uncovering these intrinsic differences in LLM behavior while also providing an opportunity to explore an under-studied area.

## 2.2 Sparse Autoencoders as a Key Analytical Tool

Sparse autoencoders (SAEs) have become a prominent method for sparse dictionary learning, aiming to break down LLM embeddings into sparse, mono-semantic features (Huben et al., 2024). While they have demonstrated remarkable success in identifying ground truth features in controlled, toy experiment settings (Sharkey & Beren, 2022), applying them to real LLMs presents several challenges. Despite these hurdles, SAEs remain a widely used baseline in mechanistic interpretability due to their simplicity and effectiveness (Lieberum et al., 2024).

SAEs, consisting of an encoder and a decoder, use a squared error reconstruction loss and a sparsity penalty:

$$L_{SAE} = \frac{1}{N} \sum_{i=1}^{N} \|\mathbf{x}^{(i)} - \hat{\mathbf{x}}^{(i)}\|_2^2 + \lambda \cdot L_{sparsity} \tag{1}$$

where $\mathbf{x}^{(i)}$ represents the original LLM dense embedding, and $\hat{\mathbf{x}}^{(i)}$ is the reconstructed output of the SAE:

$$\hat{\mathbf{x}}^{(i)} = W_{dec}\big(\sigma(W_{enc}(\mathbf{x}^{(i)}))\big) \tag{2}$$

For the activation function $\sigma$, various techniques are used, such as JumpReLU (Lieberum et al., 2024) and TopK-ReLU (Gao et al., 2024). Each column of $W_{dec}$ is designed to capture sparse, mono-semantic concepts, which we refer to as SAE features throughout this paper.

In this study, we leverage SAEs to investigate multilingualism in LLMs, uncovering several intriguing patterns. To the best of our knowledge, this is one of the first works to apply SAEs in the study of multilingualism.

# 3 Analyzing Multilingualism in LLMs with SAEs

## 3.1 Motivation

Inspired by the impressive performance of sparse autoencoders (SAEs) in mechanistic interpretability, researchers have recently begun applying them to a range of practical tasks

beyond interpretability (Demircan et al., 2024; Wu et al., 2025; Kantamneni et al., 2025). Building on this momentum, we explore their potential in the study of multilinguality. In this work, we demonstrate that SAEs can indeed yield novel insights into multilingual processing as well.

### 3.2 Our Objective and Approach

Our primary goal is to gain deeper insights into how LLMs process multilingualism. To achieve this, we take a straightforward approach by analyzing SAE-derived patterns across parallel multilingual corpora, focusing on identifying both consistent and distinct patterns between high- and low-performing languages. Ultimately, we aim to leverage these findings to improve performance in low-performing languages.

### 3.3 Proposed Methods

In this work, we present a series of findings and discussions on intriguing SAE patterns across languages. We begin by defining two key types of SAE features: Task Instruction-Focused (TF) and Heading-Focused (HF) features. Our analysis reveals a consistent discrepancy between high- and low-performing languages with respect to these feature types. Building on these findings, we propose two novel approaches for enhancing multilingual performance: English Heading Prompts (EHP) and Amplifying TF features (ATF).

We start with formally defining TF and HF features that we consider particularly interesting, which serve as the building blocks for our analysis.

**Definition of Task Instruction-Focused Features (TF Features)**   We identify several features that primarily activate on the task instruction segment and are largely deactivated in other parts of the prompt. We define these as task instruction-focused features (TF features). Specifically, a feature is considered a TF-feature if the proportion of its activated tokens within the task instruction exceeds $p$% of all its activated tokens, computed over the ARC-Challenge training set (approximately 1,000 examples). Formally, features that meet the following criterion are defined as TF features:

$$\forall f_t \quad \text{s.t.} \quad \mathbb{E}_X \left[ \frac{\sum_{i=1}^{x_T} \mathbb{1}[SAE_{Enc}(X_i)_t > 0]}{\sum_{i=1}^{x_N} \mathbb{1}[SAE_{Enc}(X_i)_t > 0]} \right] > p \tag{3}$$

Where $t$ is the feature index, $x_N$ denotes the total number of tokens, $x_T$ indicates the number of task instruction tokens, $SAE_{Enc}$ denotes the SAE encoder, $SAE_{Enc}(X_i)_t$ represents the activation of feature $t$ on the $i$-th token, and $\mathbb{1}$ represents the indicator function. We set $p = 0.8$ as the default value. An example of an actual TF-feature is shown in Figure 2.

**Definition of Heading-Focused Features (HF-Features)**   Another notable type of SAE feature we observe is those that activate primarily on headings (see Figure 3 for an example). We define these as heading-focused features (HF features) using a similar definition:

$$\forall f_t \quad \text{s.t.} \quad \mathbb{E}_X \left[ \frac{\sum_{i \in H} \mathbb{1}[SAE_{Enc}(X_i)_t > 0]}{\sum_{i=1}^{x_N} \mathbb{1}[SAE_{Enc}(X_i)_t > 0]} \right] > p \tag{4}$$

Where $H$ represents the set of headings, which are three words: "Question", "Choices", and "Answer" in our study (See Appendix A.1 for details).

**ENGLISH HEADING PROMPTS (EHP)**   In Section 4.3, we demonstrate that headings hold a unique significance for LLMs, evidenced by the existence of multiple SAE features that activate primarily on headings only (i.e., HF features). This finding suggests that headings play a crucial role in LLM processing, leading us to hypothesize that *converting headings to English—the primary pivot language for these models—could enhance performance in low-performing languages.* Notably, this approach is highly efficient because headings are typically fixed and do not require domain-specific knowledge, eliminating the need to modify or introduce new vocabulary for different test queries. As a result, our English

Heading Prompt (EHP) can be directly applied across various domains. This simple idea consistently improves performance in both MMLU and ARC especially for low-performing languages. Furthermore, EHP outperforms using an entire English task instruction, further validating the valuable role that headings play in LLMs.

**AMPLIFYING TF FEATURES (ATF)**   Several recent studies have highlighted the importance of high-quality task instructions (Srivastava et al., 2024), suggesting that task instructions could be crucial for LLM performance. Building on these findings, as well as our new insights in Section 4.2 that low-performing languages typically lack TF features, we propose Amplifying TF Features (ATF), a method that activates TF features derived from the English training set. Formally,

$$\text{ATF:} \quad \forall i \ \hat{X}_i = X_i + \alpha \cdot \sum_{t \in D_{TF}} S_D(t) \tag{5}$$

Here, $X$ represents the original embedding, $D_{TF}$ denotes the set of chosen English TF features, and $S_D(t)$ refers to the SAE decoder's $t$-th column (or feature). $D_{TF}$ and $\alpha$ are tuned on the validation set.

We demonstrate that ATF can further enhance performance, even for high-resource languages, highlighting the critical role of a thorough understanding of task instructions in achieving superior performance.

## 3.4   Experimental Settings

**Datasets**   We utilize multilingual versions of two widely used multiple-choice question-answering benchmarks: multilingual-MMLU (STEM) (OpenAI, 2024) and multilingual-ARC- Challenge (Lai et al., 2023). MMLU (STEM) assesses knowledge across diverse subject areas in STEM, while ARC focuses on reasoning based on grade-school science knowledge, assembled to encourage research in advanced question-answering. Both are formed as four-option multiple-choice questions, following their original format.

**Models and SAEs**   A key requirement of our study is access to well-trained SAEs. Fortunately, Lieberum et al. (2024) and He et al. (2024) provide suites of pretrained SAEs for the Gemma2 and Llama3.1 models. Accordingly, we use the Gemma2-2B-IT, Gemma2-9B-IT, and Llama3.1-8B-Instruct models along with their corresponding SAEs in our experiments. Unless otherwise stated, our primary analysis focuses on the middle-layer SAE.

**Experimental Details**   As discussed in the Introduction, we adopt a zero-shot setting, which remains relatively underexplored but provides a clearer lens into the fundamental differences in how LLMs process various languages. The baseline prompt used in our experiments—comprising a task instruction, headings, and a test query—is detailed in Appendix A.1. We employ greedy decoding and accuracy as our primary evaluation metric.

For our language selection, we consider total of ten languages. To ensure sufficient comparability, we selected languages present in both m-MMLU and m-ARC (e.g., languages appearing only in one dataset, such as Swahili, were excluded). Our final selection includes English (En), German (De), Chinese (Zh), Spanish (Es), French (Fr), Portuguese (Pt), Indonesian (Id), Arabic (Ar), Hindi (Hi), and Bengali (Bn). They are roughly grouped into two, high- and low-performing language groups, based on their downstream performance.

## 4   Main Findings and Discussions

### 4.1   SAEs Provide Tangible Evidence that LLMs Have a Precise Understanding of the Prompt Structure

Large language models (LLMs) have demonstrated notable performance on a wide range of tasks, which suggest that they possess a robust understanding of input prompts. However, their sensitivity to even minor prompt modifications (Shivagunde et al., 2024; Cho et al.,

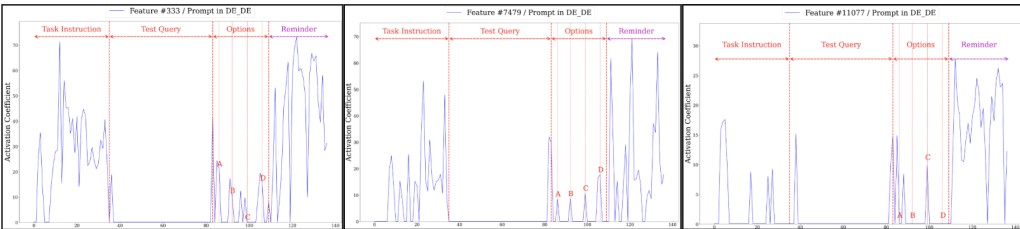

Figure 1: **Can SAEs Effectively Differentiate Between Example and Non-Example Segments in More Challenging Inputs? (We use Gemma2-9B-IT in this experiment).** We intentionally added a reminder sentence at the end to assess whether the features can dynamically toggle (i.e., activate, deactivate, and then reactivate) for non-example segments, thereby demonstrating their flexibility. The prompt consists, in order, of a task instruction, a test example query, choice options, and a reminder. The figure above shows that multiple features activate in both the initial and final segments, while the test-query portion in the middle remains strictly deactivated. These results provide compelling evidence that LLMs have a precise awareness of the structure of the prompt (e.g., task-instruction-related parts and the test-query components).

2024) raises important questions about what it truly means for an LLM to "understand" a prompt—and whether we can obtain more fine-grained evidence of this comprehension beyond downstream task performance (Cho et al., 2025).

In this work, we demonstrate that sparse autoencoders (SAEs) provide compelling evidence that LLMs possess a precise understanding of distinct prompt segments. Our findings reveal that LLMs can effectively differentiate among task instructions, main example (i.e., test query), and headings. Figures 2 and 3 illustrate this phenomenon: Figure 3 (b) shows that Feature #14428, classified as an HF feature by our definition, activates primarily on headings across all examined languages, while the bottom of Figure 2 shows that Feature #9349 fires predominantly on task instructions across all studied languages (i.e., it is a TF feature). The presence of these features serves as strong, tangible evidence that LLMs possess an accurate understanding of prompt structure, and that comprehension is, to some extent, universally shared across languages.

To further evaluate LLMs' ability to distinguish prompt segments, we conducted a study in which we added a sentence irrelevant to the main example (i.e., the test query). Specifically, we inserted a "Reminder" sentence—"Make sure your answer is one of A, B, C, or D."—at the end of the prompt. If the LLM activates features on this non-example segment, along with the previous non-example segments before the test query, it provides stronger evidence of its precision in distinguishing between main example and non-main example segments. In other words, we aim to assess whether the features can dynamically toggle (on, off, and then on again), demonstrating their flexibility. As shown in Fig. 1, Features #333, #7479, #11077, among others (using Gemma2-9B-IT), effectively differentiate these segments. The existence of these highly accurate segment-aware features offers valuable insights into how LLMs truly understand the prompt.

## 4.2 Low-performing Languages Relatively Lack Task Instruction-Focused Features

An intriguing type of feature we observe is the task instruction-focused (TF) features, as defined in Section 3.3. To analyze these features, we first compute their statistics and compare the number of TF features across different languages. Specifically, we process all tokens in the ARC training set through the SAE encoder and collect the top 500 most frequently activated features. This top-500 selection ensures that we focus on the most meaningful features, filtering out those that activate only sporadically (e.g., once in every thousand tokens), which are likely to be less significant. Then, we compute the number of TF features (p=0.8) and the results are shown at the top of Figure 2.

Overall, we observe three key findings regarding TF features:

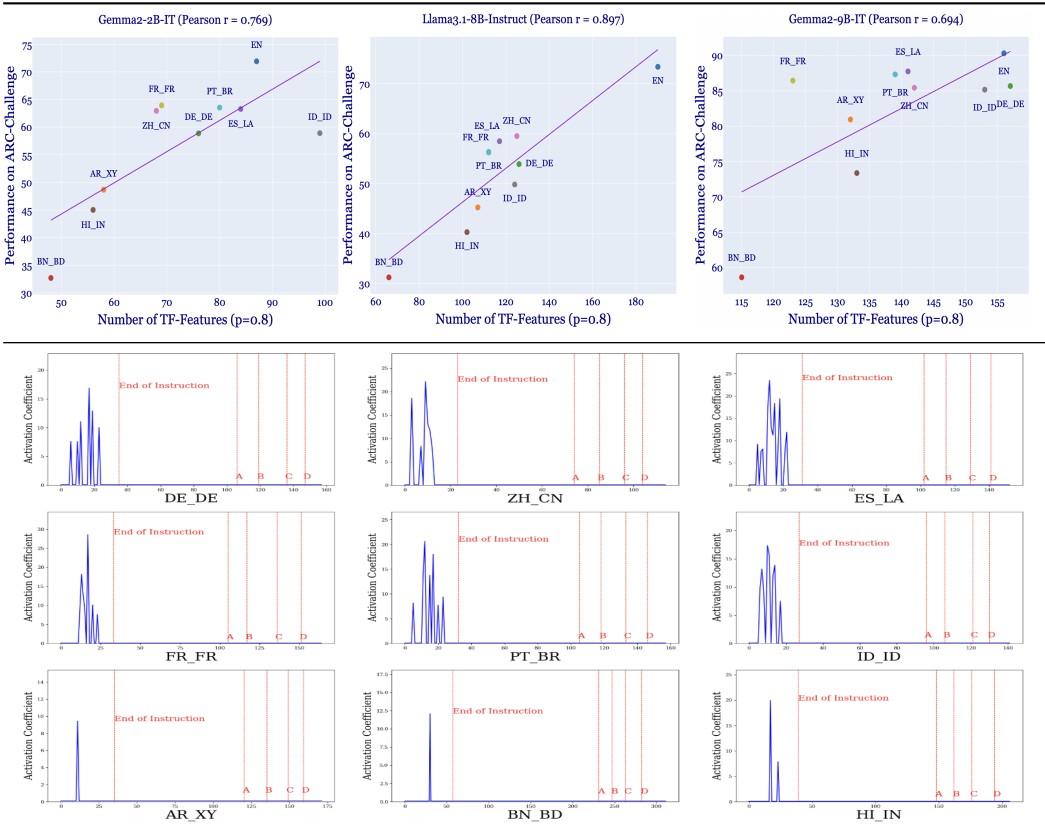

Figure 2: **Existence of Task Instruction-Focused (TF) Features and Their Correlation with Language Performance:** (a) The first three plots at the top row show a strong correlation between language performance and the number of TF features. This consistent trend across three different models—Gemma2-2B-IT, Llama3.1-8B-Instruct, and Gemma2-9B-IT—suggests that TF features can provide new insights into the causes of performance gaps between languages in LLMs. (b) (Feature #9349 in Gemma2-9B-IT): Although many TF features are multilingual, there is often a significant difference in their activation frequency: notably, low-performing languages (as shown in the bottom row) have fewer tokens activating these features.

**TF Features Shed Light on LLM Prompt Processing**   SAEs are trained in a completely unsupervised, bag-of-embeddings fashion, with no inherent expectation that TF features would emerge. Thus, their presence—and especially their abundance—is both surprising and significant, as it offers concrete, measurable proof that LLMs can differentiate between task instruction segments and non-task instruction segments within a prompt. This finding provides valuable insights into how LLMs break down and process prompts, and paves the way for promising future research directions.

**TF Features Provide Deeper Insights into Language Discrepancies**   As illustrated in Figure 2 (a), the number of TF features differs significantly between high- and low-performing languages. Additionally, the number of TF features exhibits a decent correlation with overall downstream performance (see Figure 2), suggesting that these features could play a crucial role in language performance.

**TF Features Are Generally Universal, but Their Activation Frequency Is Lower for Low-Performing Languages**   The bottom of Figure 2 shows that although Feature #9349 in Gemma2-9B-IT's SAE activates across all languages, low-performing languages exhibit a lower activation frequency (as indicated in the last row), highlighting subtle discrepancies

between high- and low-performing languages (more examples in Appendix A.1). We argue that these consistent patterns—wherein low-performing languages lack TF features in both number and frequency—indicate that TF features play a crucial role in LLMs' multilingual performance.

Overall, the key takeaway is that TF features are strong candidates for playing a crucial role in the multilingual performance of LLMs.

### 4.3 The Importance of Headings in LLMs

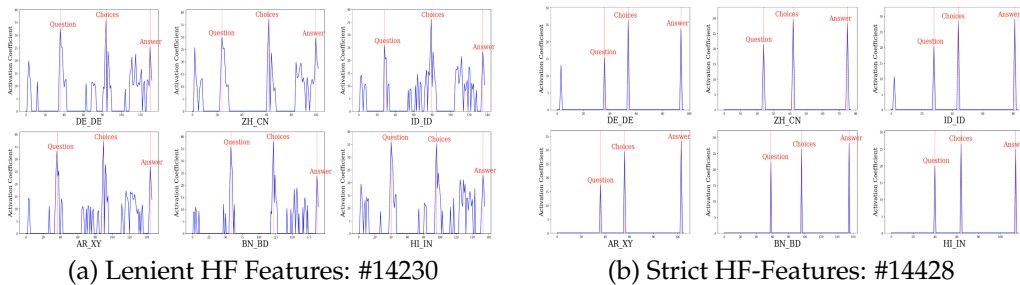

(a) Lenient HF Features: #14230        (b) Strict HF-Features: #14428

Figure 3: **Existence of Various HF features (using Gemma2-9B-IT).** The abundance of these features serves as evidence that headings hold a special significance for LLMs compared to other general words, indicating that LLMs have a clear understanding and focus on headings when processing the prompt. The figures above suggest that HF features can be further classified into lenient and strict HF features.

Another intriguing class of features we identify is the heading-focused (HF) features, as defined in Section 3.3. Similar to TF features, the emergence of HF features is surprising, as there is no inherent guarantee that they should exist. For example, as shown in Figure 3(b), feature #14428 (using Gemma2-9B-IT) consistently activates on headings across multiple languages. The abundant existence of HF features provide clear evidence that LLMs possess a sophisticated understanding of headings—a critical component of prompt structure.

Another noteworthy observation is the presence of somewhat "noisy" HF features. For instance, Figure 3 (a) shows Feature #14230, which is clearly heading-aware, as it activates strongly on headings, though not as precisely as Feature #14428 in Figure 3 (b). Given that no model is perfect, such imperfections in SAEs can be understood. Nevertheless, this suggests that there could be many more HF features than our measurements if we also account for the lenient ones. Investigating TF and HF features in more detail would be an interesting area for future work.

### 4.4 Enhancing Low-performing Language Performance: English Heading Prompts (EHP)

Based on our findings, we have identified multiple HF features, indicating that LLMs place significant emphasis on headings, which play a unique role in their mechanism. Inspired by this observation, we introduce English Heading Prompts (EHP), a method that replaces headings with English words. This approach is motivated by the fact that LLMs are predominantly English-centric and use English as the pivot language, enabling them to more effectively leverage the critical role of headings when presented in English rather than in low-performing languages.

The results, summarized in Table 1, show that EHP consistently improves performance across languages, with a relatively larger impact on low-performing languages. These findings empirically validate our hypothesis that headings play a crucial role in LLM operations. What is especially noteworthy is that EHP outperforms the use of a complete English task instruction (denoted as ETI in Table 2). Despite ETI generally containing a larger word count in English, EHP proves to be more effective, further underscoring the important role headings play in LLMs.

| | | | | | | | | | | | | |
|---|---|---|---|---|---|---|---|---|---|---|---|---|
| | | | | Task: Multilingual ARC-Challenge | | | | | | | | |
| | | | | HPL | | | | | LPL | | Avg. | |
| *Gemma2-9B-IT* | En | De | Zh | Es | Fr | Pt | Id | Ar | Hi | Bn | HPL Avg. | LPL Avg. |
| Zero-shot (ZS) | 90.30 | 84.98 | 85.08 | 87.05 | 85.75 | 85.76 | 83.45 | 78.54 | 72.34 | 57.68 | 85.72 | 73.00 |
| ZS + EHP | 90.30 | 85.24 | 85.16 | 87.31 | 85.75 | 86.11 | 85.08 | 79.31 | 73.11 | 58.80 | 85.93 | 74.08 |
| **ZS + EHP + ATF** | **90.82** | **85.84** | **85.76** | **87.65** | **86.27** | **86.88** | **85.51** | **80.43** | **73.54** | **59.91** | **86.48** | **74.85** |
| | | | | | | | | | | | | |
| *Gemma2-2B-IT* | En | De | Zh | Es | Fr | Pt | Id | Ar | Hi | Bn | HPL Avg. | LPL Avg. |
| Zero-shot (ZS) | 71.93 | 58.11 | 62.01 | 64.41 | 65.06 | 63.04 | 60.12 | 50.30 | 48.11 | 33.73 | 62.12 | 44.05 |
| ZS + EHP | 71.93 | 59.48 | 61.23 | 64.84 | 64.38 | 63.12 | 60.38 | 50.56 | 48.97 | 34.94 | 62.24 | 44.82 |
| **ZS + EHP + ATF** | **72.45** | **60.09** | **62.52** | **65.52** | **64.89** | **64.15** | **61.58** | **51.59** | **49.91** | **36.48** | **63.12** | **45.99** |
| | | | | | | | | | | | | |
| *Llama3.1-8B-Instruct* | En | De | Zh | Es | Fr | Pt | Id | Ar | Hi | Bn | HPL Avg. | LPL Avg. |
| Zero-shot (ZS) | 72.45 | 53.91 | 59.52 | 58.49 | 56.22 | 56.35 | 49.83 | 45.24 | 40.29 | 31.24 | 55.72 | 38.92 |
| ZS + EHP | 72.45 | 55.88 | 59.26 | 59.52 | 56.65 | 57.63 | 51.20 | 45.67 | 40.72 | 30.99 | 56.69 | 39.13 |
| **ZS + EHP + ATF** | **73.91** | **56.22** | **59.52** | **60.46** | **57.08** | **58.92** | **51.50** | **46.35** | **41.58** | **31.76** | **57.28** | **39.90** |

| | | | | | | | | | | | | |
|---|---|---|---|---|---|---|---|---|---|---|---|---|
| | | | | Task: Multilingual-MMLU (STEM) | | | | | | | | |
| Gemma2-9B-IT | | | | HPL | | | | | LPL | | Avg. | |
| | En | De | Zh | Es | Fr | Pt | Id | Ar | Hi | Bn | HRL Avg. | LRL Avg. |
| Zero-shot (ZS) | 63.72 | 57.04 | 54.93 | 59.94 | 58.37 | 56.26 | 53.05 | 48.04 | 51.02 | 46.17 | 57.31 | 49.57 |
| ZS + EHP | 63.72 | 58.92 | 56.73 | 60.17 | 59.31 | 58.84 | 57.04 | 50.86 | 53.05 | 48.67 | 58.79 | 52.40 |
| ZS + EHP+ATF | **63.95** | **60.25** | **58.22** | **60.49** | **59.75** | **59.15** | **57.12** | **52.90** | **53.83** | **49.22** | **59.57** | **53.27** |
| | | | | | | | | | | | | |
| *Gemma2-2B-IT* | En | De | Zh | Es | Fr | Pt | Id | Ar | Hi | Bn | HPL Avg. | LPL Avg. |
| Zero-shot (ZS) | 47.18 | 39.56 | 40.32 | 41.25 | 40.29 | 40.39 | 39.60 | 35.49 | 35.22 | 30.78 | 40.23 | 33.83 |
| ZS + EHP | 47.18 | 40.32 | 40.95 | 41.62 | 40.49 | 41.02 | 39.30 | 36.28 | 35.55 | 31.91 | 40.62 | 34.58 |
| **ZS + EHP + ATF** | **47.71** | **40.95** | **41.82** | **42.28** | **41.98** | **41.72** | **40.15** | **36.78** | **36.35** | **33.53** | **41.48** | **35.55** |
| | | | | | | | | | | | | |
| *Llama3.1-8B-Instruct* | En | De | Zh | Es | Fr | Pt | Id | Ar | Hi | Bn | HPL Avg. | LPL Avg. |
| Zero-shot (ZS) | 48.71 | 39.50 | 40.62 | 41.82 | 40.29 | 40.39 | 38.44 | 34.86 | 34.29 | 30.78 | 40.18 | 33.31 |
| ZS + EHP | 48.71 | 39.43 | 41.95 | 41.88 | 40.49 | 41.02 | 38.30 | 35.39 | 34.26 | 31.91 | 40.51 | 33.85 |
| **ZS + EHP + ATF** | **48.71** | **39.76** | **42.28** | **42.41** | **40.49** | **41.58** | **38.73** | **36.28** | **35.49** | **33.13** | **40.87** | **34.97** |

Table 1: **Overall results of our proposed approaches.** We observe that our methods consistently improve performance across both high-performing (HPL) and low-performing (LPL) languages.

## 4.5 Enhancing Multilingual Performance Via Amplifying TF Features (ATF)

In the previous sections, we discovered the existence of multiple TF features and observed distinct behaviors across high- and low-performing languages, along with a strong correlation between the TF feature ratio and overall task performance. Motivated by these findings, we introduce a new methodology, "Amplifying TF Features (ATF)" as defined in Section 3.3. In this approach, for each language, we first derive TF features from the English ARC-Challenge training set using equation (3). We then randomly select a subset of these TF features for steering. The number of selected features $|D_{TF}|$, the specific TF features chosen, and the value of $\alpha$ are all tuned on the validation set of the target language.

The results in Table 1 indicate that ATF is effective for both low- and high-performing languages. We believe these findings provide concrete evidence that TF features play a crucial role in LLMs' performance, an intriguing finding that offers a novel insight into the multilingual capabilities of LLMs.

## 4.6 An Additional Interesting Type of SAE Features Not Covered in This Work, but Worth Noting

We also identify features that may contribute to label bias in LLMs. For instance, Feature #4306 predominantly activates on headings and alphabetic answer choices but consistently

| Model: Gemma2-9B-IT | Task: Multilingual ARC-Challenge | | | | | | | | | |
|---|---|---|---|---|---|---|---|---|---|---|
| | HPL | | | | | | LPL | | | Avg. |
| | De | Zh | Es | Fr | Pt | Id | Ar | Hi | Bn | HPL Avg. | LPL Avg. |
| Zero-shot (ZS) | 84.98 | 85.08 | 87.05 | 85.75 | 85.76 | 83.45 | 78.54 | 72.34 | 57.68 | 85.72 | 73.00 |
| ZS + EHP | 85.24 | 85.16 | 87.31 | 85.75 | 86.11 | 85.08 | 79.31 | 73.11 | 58.80 | 85.93 | 74.08 |
| Abl 1) ZS + ETI | 84.64 | 84.73 | 86.54 | 85.75 | 86.22 | 83.79 | 79.14 | 71.74 | 57.94 | 85.58 | 73.15 |
| Abl 2) ZS + EHP + ETI | 84.64 | 85.16 | 86.79 | 85.84 | 86.22 | 84.22 | 79.14 | 72.68 | 58.11 | 85.73 | 73.54 |
| | Task: Multilingual-MMLU (STEM) | | | | | | | | | |
| | HPL | | | | | | LPL | | | Avg. |
| | De | Zh | Es | Fr | Pt | Id | Ar | Hi | Bn | HPL Avg. | LPL Avg. |
| Zero-shot (ZS) | 57.04 | 54.93 | 59.94 | 58.37 | 56.26 | 53.05 | 48.04 | 51.02 | 46.17 | 57.31 | 49.57 |
| ZS + EHP | 58.92 | 56.73 | 60.17 | 59.31 | 58.84 | 57.04 | 50.86 | 53.05 | 48.67 | 58.79 | 52.40 |
| Abl 1) ZS + ETI | 58.84 | 57.43 | 59.62 | 58.14 | 58.76 | 55.01 | 50.94 | 50.55 | 46.64 | 58.56 | 50.79 |
| Abl 2) ZS + EHP + ETI | 58.84 | 57.20 | 60.09 | 59.23 | 59.23 | 55.87 | 51.49 | 52.58 | 45.85 | 58.92 | 51.45 |

Table 2: **Ablation results on the distinctiveness of headings (using Gemma2-9B-IT).** Using English headings in the prompt generally improves performance, and interestingly, this is often more effective than using the entire English task instruction—indicating that headings play a unique and powerful role in LLMs.

shows zero activation for 'A' alone (see Figure 6), regardless of the specific example. We believe this bias could contribute to the well-known label-bias issue in LLMs. Suppressing these features by steering may help mitigate this problem, and we leave this promising direction for future work.

## 5 Conclusion

This paper investigates whether sparse autoencoders (SAEs) can enhance our understanding of how large language models (LLMs) handle multilingualism. We introduce two novel types of SAE features: Task Instruction-Focused (TF) features and Heading-Focused (HF) features. Our analysis reveals several novel insights: (1) TF and HF features shed light on how LLMs parse and process prompt structures; (2) headings play a particularly important role in prompts, as evidenced by the abundance of HF-features; and (3) TF features are crucial for multilingual performance. Based on these findings, we propose two practical approaches to enhance multilingual performance: English Heading Prompts and Amplifying TF Features, which consistently enhances performance, especially for low-performing languages. To the best of our knowledge, this work is among the first to explore the versatility of SAEs in multilingualism, highlighting their potential and offering fresh insights into their underlying mechanisms.

## 6 Acknowledgements

This research was supported in part by Other Transaction award HR0011249XXX from the U.S. Defense Advanced Research Projects Agency (DARPA) Friction for Accountability in Conversational Transactions (FACT) program. Additionally, this research used the Delta advanced computing and data resource which is supported by the National Science Foundation (award OAC 2005572) and the State of Illinois. Delta is a joint effort of the University of Illinois Urbana-Champaign and its National Center for Supercomputing Applications.

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

## A  Appendix

### A.1  Prompts Used in Our Study

**Baseline Prompt**

[Task Instruction in Language X]
["Question" in Language X]:
[Test Query]
["Choices" in Language X]:
A:
[Option A in Language X]
B:
[Option B in Language X]
C:
[Option C in Language X]
D:
[Option D in Language X]
["Answer" in Language X]:

Figure 4: **Baseline Prompt.** The baseline prompt used in our study. Headings are highlighted in blue.

The baseline prompt we use in our study is shown in Figure 4 (a). The headings are highlighted in blue, and each square bracket pair indicates a placeholder based on the test query and language.

### A.2  More Examples of SAE Features

In Section 4.2, we noted that even when a TF-feature is multilingual, the frequency of activated tokens is generally lower in low-resource languages, indicating a clear distinction between high- and low-resource languages. Here, we present another such feature in Figure 5 along with corresponding statistics using the ARC dataset.

In Section 4.6, we introduced another potentially significant feature type that may contribute to label bias. Examples of these features are illustrated in Figure 6.

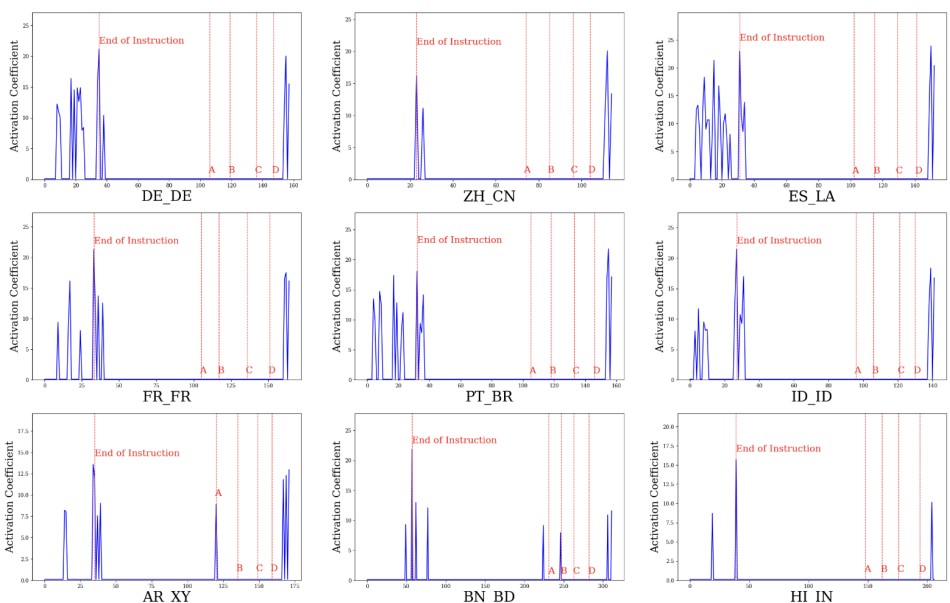

Figure 5: **Decline in TF Feature Frequency for Low-performing Languages.** We provide another example of TF-features showing a significant reduction in activation frequency for low-resource languages (see the last row).

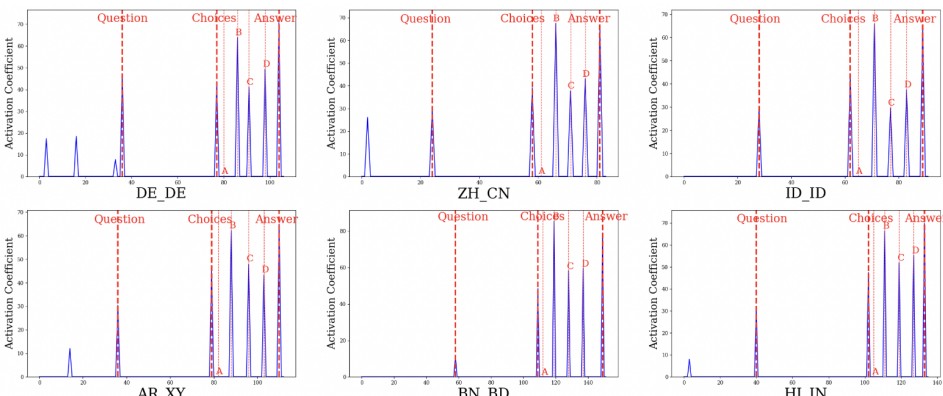

Figure 6: **Features that may contribute to label bias:** Feature #4306 exhibits behavior similar to HF-features, as it strongly activates on headings. However, we observe a consistent pattern where its activation for 'A' is significantly lower than for 'B,' 'C,' and 'D' across most examples.

