# OpenReview forum: "Analyzing Multilingualism in Large Language Models with Sparse Autoencoders"
_colmweb.org/COLM/2025/Conference — COLM 2025_

### Official Review · Reviewer_Bq9B · 2025-05-12

**Rating:** 6
**Confidence:** 4
**Ethics Flag:** 1

**Summary:**

The paper investigates the Gemma 2 9b model using sparse autoencoder (SAE) features from the perspective of multilinguality via multilingual multiple choice tasks (m-MMLU-STEM and m-ARC).
The paper argues that the presence of 'active' SAE features correlate well with the prompt structure.
Specifically, the authors distinguish Task Instruction-focused (TF) and Heading-focused (HF) features.
The presence of such features are contrasted between low and high resource languages.
As TF-features are less frequent for lower-resource languages, the paper also proposes a technique for their amplification.

**Questions To Authors:**

How exactly were the languages classified into low and high resource languages?

Comments:
The bar for ZH_CN is absent from Figure 2 (a).
In §3.1 (lines 130--132), it was strange to see (Kantamneni et al., 2025) being cited for supporting the ``remarkable performance'' of SAEs, whereas that paper is rather critical towards the usefulness of SAEs.

**Reasons To Accept:**

Investigating the multilingual capabilities of LLMs is a timely topic, for which a simple solution is provided in the paper.

**Reasons To Reject:**

There are certain design choices in the experimental protocol that are not well justified, with a potentially strong effect on the core findings of the paper.
For instance, the definition of TF (and HF) features have the limitation that it can assign high scores for SAE features that are active very infrequently.
There is a heuristic implemented in §4.2 (i.e. considering only the 500 most frequently activated features) to mitigate the problem, however, there was an easy and more principled fix of considering the (normalized) pointwise mutual information between the presence of a certain property (a token being task-instruction focused) and a particular SAE feature being active as it have already been proposed for sparse contextualized representations in prior work.
The threshold for selecting a SAE feature is not justified either, nor it is discussed why the MMLU dataset was constrained for STEM-related questions.
Additionally, the classification of the tested languages into low and high resource is lacking, which is problematic, as certain findings are based on that distinction.
Finally, it would contextualize the results better if results for English were also reported, as this could help in seeing the contrastive difference of the behavior of the SAE features between the language the model was primarily created for and additional target languages.

---

> ### Author Response · Authors · 2025-06-03
>
> We sincerely appreciate the reviewer’s thoughtful feedback. Below, we provide additional experimental results and clarifications to address your concerns.
>
> **1. Frequency as an Initial Filter.**
>
> We agree that there are alternative ways to define TF features. We adopted frequency-based filtering as an initial step to exclude features that activate on too few tokens, primarily for its simplicity. We found this approach sufficient for effectively identifying TF features—for instance, Figure 1 in the paper shows that the resulting TF features consistently activate on task-instruction-related segments. In line with your suggestion, we agree that exploring alternative definitions and comparing their effects would offer valuable insights, and we plan to pursue this during the remaining week of the rebuttal period.
>
> **2. Threshold.**
>
> The primary contribution of our work is to demonstrate that TF and HF features exist in SAEs, with meaningful differences in their ratios across languages. These definitions are most meaningful when the threshold parameter p is close to 1, aligning with their intended motivations. In response to your comment, we examined a range of p values and found that the overall patterns remain stable (i.e., high-performing languages consistently exhibit higher TF-feature ratios than low-performing languages) under small perturbations around p = 1. The corresponding statistics are provided in Table 1 (We have also updated the terminology to “High-Performing Languages (HPL)” and “Low-Performing Languages (LPL),” and added two additional models—Gemma2-2B-IT and Llama 3.1-8B-Instruct—with more detailed explanation provided in #6 below.)
>
> |TF-Feature Ratio (%)||Bn|Ar|Hi|Id|De|Es|Fr|Pt|Zh|\| En|HPL AVG|LPL AVG|
> |-|:-:|:-:|:-:|:-:|:-:|:-:|:-:|:-:|:-:|:-:|:-:|:-:|:-:|
> |*Llama3.1-8B-Instruct*|
> |* p=0.9||5.21|14.80|10.59|13.29|18.73|18.17|16.58|17.09|22.86|**\|** 24.87|18.69|10.97|
> |* p=0.8||6.14|15.38|11.27|14.02|19.46|18.91|18.12|17.54|23.11|**\|** 25.56|19.43|11.70|
> |* p=0.7||8.85|17.74|13.46|16.48|21.33|21.57|19.95|20.99|24.26|**\|** 27.50|21.62|14.13|
> |* p=0.6||13.32|23.36|19.29|22.06|25.44|25.67|23.79|25.01|27.15|**\|** 30.16|25.41|19.51|
> |* p=0.5||17.34|25.03|21.79|24.25|27.06|27.68|25.13|26.43|28.00|**\|** 31.05|26.86|22.10|
> |* p=0.4||26.68|39.47|34.28|38.04|39.60|39.72|37.82|38.94|37.80|**\|** 42.93|38.78|34.62|
> |*Gemma2-2B-IT*|
> |* p=0.9||0.43|0.78|0.66|1.81|5.16|4.04|3.51|3.46|3.13|**\|** 5.50|3.86|0.92|
> |* p=0.8||1.39|2.82|2.58|4.86|8.71|7.07|6.56|6.20|5.70|**\|** 9.29|6.85|2.91|
> |* p=0.7||3.56|4.97|5.55|8.07|12.12|10.25|10.06|9.42|9.37|**\|** 12.25|10.24|5.54|
> |* p=0.6||7.38|9.90|9.18|13.40|16.51|14.77|13.71|13.64|14.54|**\|** 16.39|14.63|9.96|
> |* p=0.5||14.60|16.96|15.76|20.85|22.78|21.37|19.87|20.07|21.07|**\|** 22.82|21.03|17.04|
> |* p=0.4||25.59|24.68|23.16|27.26|29.09|27.36|25.99|26.42|26.96|**\|** 28.88|27.16|25.17|
> ||
>
> *Table 1: Effect of the hyperparameter p on TF and HF features. Note that the definitions align with their intended motivation when p is close to 1.*
>
> **3. MMLU-STEM.**
>
> The main reason for restricting our evaluation to MMLU-STEM topics was the large size of the full MMLU dataset (approximately 15k examples). We selected a smaller STEM-focused subset (around 1.3k examples), which is comparable in size to the ARC-Challenge dataset (1.2k), to maintain experimental feasibility.
>
> **4. Language Classification.**
>
> Thank you for pointing this out. We initially categorized the languages primarily based on their task performance, supplemented by some preliminary knowledge of their presence in the pretraining data. Given your concern—and the fact that the exact distribution of languages in the model’s pretraining and instruction tuning data is difficult to determine—we are happy to revise the terminology to 'high-performing' and 'low-performing' languages with respect to a specific task.
>
> **5. Presentation  Issues.**
>
> Thank you for bringing this to our attention. We will include the bar for ZH_CN in the plot (the corresponding value can be found in Table 2 of the Appendix). Also, regarding the reference, our intention was to cite it as an example of applying SAEs in practical tasks, but we agree it may appear out of place. We will revise it accordingly in the updated version.
>
> **6. Additional Experiments Including English Results.**
>
> (Continued)

---

> > ### Author Response · Authors · 2025-06-03
> >
> > (Continued)
> >
> > **6. Additional Experiments Including English Results.**
> >
> > Regarding your feedback, we have added two additional models—Gemma2-2B-IT and Llama 3.1-8B-IT (using SAEs from layer 20 with width 65k from the Gemma scope, and layer 23 with width 131k from the Llama scope)—to assess the generalizability of our approach. We have also included English results in the tables below. We’re excited to share these encouraging findings.
> > |Llama3.1-8B-IT||Bn|Ar|Hi|Id|De|Es|Fr|Pt|Zh|\| En|HPL AVG|LPL AVG|
> > |-|:-:|:-:|:-:|:-:|:-:|:-:|:-:|:-:|:-:|:-:|:-:|:-:|:-:|
> > |TF-Feature Ratio (%)||5.21|14.80|10.59|13.29|18.73|18.17|16.58|17.09|22.86|**\|** 24.87|18.69|10.97|
> > |HF-Feature Ratio (%)||0.10|0.56|0.35|1.44|1.29|0.67|0.31|0.66|1.06|**\|** 1.41|0.80|0.61|
> > |**Gemma2-2B-IT**|
> > |TF-Feature Ratio (%)||0.43|0.78|0.66|1.81|5.16|4.04|3.51|3.46|3.13|**\|** 5.50|3.86|0.92|
> > |HF-Feature Ratio (%)||0.01|0.17|0.22|1.04|1.09|0.67|1.22|0.98|1.02|**\|** 1.19|1.00|0.36|
> > ||
> >
> > *Table 2: HPL exhibit higher TF and HF feature ratio than LPL on both models.*
> >
> > | *Gemma2-2B-IT*||||||||||||||
> > |-|:-:|:-:|:-:|:-:|:-:|:-:|:-:|:-:|:-:|:-:|-|:-:|:-:|
> > |**ARC**||Bn|Ar|Hi|Id|De|Es|Fr|Pt|Zh|\| En|HPL AVG|LPL AVG|
> > |Zero-Shot Baseline||32.70|48.67|45.02|58.92|58.88|63.29|63.95|63.55|62.95|**\|** 71.93|62.52|46.33|
> > |ETI||34.76|48.07|46.65|59.61|59.06|64.15|62.92|62.92|61.75|**\|** 71.93|62.16|47.17|
> > |EHP||33.73|49.27|46.99|59.52|59.14|64.32|63.52|64.49|63.04|**\|** 71.93|62.90|47.38|
> > |EHP+ATF||**34.76**|**50.47**|**47.85**|**60.89**|**60.00**|**65.01**|**64.55**|**64.92**|**63.52**|**\|** **72.45**|**63.60**|**48.49**|
> > ||
> > |**MMLU-STEM**||Bn|Ar|Hi|Id|De|Es|Fr|Pt|Zh|\|En|HPL AVG|LPL AVG|
> > |Zero-Shot Baseline||30.36|34.35|35.68|38.65|39.51|41.16|40.38|39.44|41.24|**\|** 50.63|40.35|34.76|
> > |ETI||31.92|34.66|35.68|38.11|40.30|40.92|40.69|39.51|39.83|**\|** 50.63|40.25|35.09|
> > |EHP||32.16|35.52|35.76|40.14|40.30|41.86|41.39|40.53|40.62|**\|** 50.63|40.94|35.89|
> > |EHP+ATF||**32.86**|**36.15**|**36.46**|**40.30**|**40.85**|**42.10**|**41.63**|**41.55**|**40.69**|**\|** **51.17**|**41.36**|**36.44**|
> > ||
> >
> > *Table 3: Results on Gemma2-2B-IT.*
> >
> > | *Llama3.1-8B-Instruct*||||||||||||||
> > |-|:-:|:-:|:-:|:-:|:-:|:-:|:-:|:-:|:-:|:-:|:-:|:-:|:-:|
> > |**ARC**||Bn|Ar|Hi|Id|De|Es|Fr|Pt|Zh|\| En|HPL AVG|LPL AVG|
> > |Zero-Shot Baseline||13.65|41.03|36.43|51.03|53.99|58.15|58.03|56.60|58.40|**\|** 72.45|57.03|35.54|
> > |ETI||10.82|43.86|34.19|49.91|53.48|57.71|56.48|55.75|59.18|**\|** 72.45|56.52|34.70|
> > |EHP||22.15|46.18|40.12|53.09|55.11|59.86|58.20|58.92|58.49|**\|** 72.45|58.12|40.38|
> > |EHP+ATF||**31.24**|**47.73**|**41.58**|**53.77**|**56.48**|**60.46**|**58.63**|**59.61**|**59.52**|**\|** **73.91**|**58.94**|**43.58**|
> > ||
> > |**MMLU-STEM**||Bn|Ar|Hi|Id|De|Es|Fr|Pt|Zh|\|En|HPL AVG|LPL AVG|
> > |Zero-Shot Baseline||16.43|32.32|32.55|36.15|40.77|43.43|42.18|41.63|41.39|**\|** 48.83|41.88|29.36|
> > |ETI||9.00|31.22|31.38|35.21|40.61|41.55|42.10|41.24|41.94|**\|** 48.83|41.49|26.70|
> > |EHP||26.29|35.13|35.05|38.89|41.24|44.52|43.04|42.57|42.18|**\|** 48.83|42.71|33.84|
> > |EHP+ATF||**30.67**|**36.62**|**37.25**|**39.91**|**42.57**|**45.62**|**44.60**|**43.51**|**43.11**|**\|** **49.84**|**43.88**|**36.11**|
> > ||
> >
> > *Table 4: Results on Llama3.1-8B-Instruct.*
> >
> > The key takeaways are as follows:
> > * The main patterns from the original paper—such as the higher TF and HF feature ratios for high-performing languages (Table 2) and the effectiveness of EHP and ATF (Tables 3 and 4)—are consistent across the new models.
> > * English, as the pivot and highest-performing language, shows the highest TF-feature ratio and the second-highest HF-feature ratio among all languages examined across both models (Table 2). Moreover, English performance can be also improved through ATF, which we believe is an encouraging result (Tables 3 and 4).
> >
> > These results indicate that our key findings from the original paper generalize across different model sizes and architectures, thereby reinforcing our main contribution—namely, that the concepts of TF and HF features provide novel and fresh insights into the origins of language performance gaps in LLMs.
> >
> > ---
> >
> > Thank you once again for your valuable feedback and for taking the time to review our work. We hope these additions have addressed your concerns and strengthened the paper. If you find the revisions satisfactory, we would deeply appreciate your reconsideration of the rating.

---

> > > ### Comment · Reviewer_Bq9B · 2025-06-10
> > > **Thank you**
> > >
> > > In light of your response I increased my initial score.
> > >
> > > I appreciate that you reconsider the naming for the language classification that you initially introduced (i.e. instead of high and low resource you would use high-performing and low-performing languages), but I do agree with Reviewer BE52 in that this introduces some level of undesirable subjectivity in the interpretation of the results.

---

> > > > ### Author Response · Authors · 2025-06-10
> > > >
> > > > Dear Reviewer,
> > > >
> > > > Thank you for your follow-up!
> > > >
> > > > Based on your (and reviewer BE52's) feedback, we’ve carefully reconsidered how to improve our presentation. Since our goal is not to classify languages into groups, but rather to provide insights into the underlying differences in their performance, we believe it is more appropriate to reframe our presentation to emphasize that the **TF-feature ratio is strongly correlated with task performance.** Indeed, using the numbers reported in our initial response, we computed correlation coefficients of 0.928 for LLaMA3.1-8B-Instruct, 0.844 for Gemma2-2B-IT—both indicating a strong relationship. We will revise the draft accordingly and hope that this new framing presents our findings more clearly and with less ambiguity.
> > > >
> > > > Thank you again for taking the time to review this material.
> > > >
> > > > Sincerely,
> > > >
> > > > The Authors

---

### Official Review · Reviewer_ZoVv · 2025-05-13

**Rating:** 6
**Confidence:** 2
**Ethics Flag:** 1

**Summary:**

This paper investigates the mechanisms underlying multilingual processing in LLMs using SAEs. The authors introduce two novel concepts based on SAE feature activations: Task Instruction-focused (TF) features, which primarily activate on task instructions, and Heading-focused (HF) features, which activate on prompt headings (like "Question", "Answer"). Their analysis, conducted on Gemma2-9B using pre-trained SAEs in a zero-shot setting on m-MMLU-STEM and m-ARC benchmarks, reveals several key findings: (1) SAEs provide evidence that LLMs have a precise understanding of prompt structure, differentiating instruction, query, and heading segments. (2) Low-resource languages exhibit a significantly lower ratio and activation frequency of TF-features compared to high-resource languages, correlating with downstream performance. (3) Headings play a distinct role, evidenced by abundant HF-features. Based on these insights, the paper proposes two methods to enhance multilingual performance, especially for low-resource languages: (a) English Heading Prompts (EHP), replacing native language headings with English ones, and (b) Amplifying TF-Features (ATF), steering the model by adding activations of identified TF-features. Combining these methods yields performance improvements of up to 3.7% on average for low-resource languages on MMLU.

**Reasons To Accept:**

- The paper introduces a novel approach by leveraging SAEs, a tool primarily used for monolingual interpretability, to gain insights into the challenging and important area of LLM multilingualism. Identifying specific feature types (TF/HF) linked to prompt structure and language resource level is a significant conceptual contribution.
- The study provides compelling evidence, grounded in SAE feature activations, for how LLMs might parse and understand prompt components differently across languages. The finding that low-resource languages have a relative deficiency in TF-features offers a potential explanation for performance gaps and is an interesting observation. The identification of dedicated HF-features highlights the structural importance of headings in prompts.
- The paper proposes two concrete methods (EHP and ATF) derived directly from the analytical findings. EHP is simple and efficient, while ATF demonstrates the potential of feature steering for multilingual improvement.

**Reasons To Reject:**

- The findings and methods are demonstrated only on one model (Gemma2-9B) and its specific pre-trained SAEs (primarily Layer 35, 16k width). It's unclear how well these findings (existence/ratio of TF/HF features) and the effectiveness of EHP/ATF generalize to other model architectures (e.g., Llama, Mistral), model sizes, other layers, or different SAE training methodologies/hyperparameters. This limitation is acknowledged but remains significant.
- The definition of TF/HF features relies on a specific activation threshold (p=0.9). The paper does not explore how sensitive the identification of these features and the subsequent analysis are to this threshold. Similarly, the ATF method involves hyperparameters (|DTF|, alpha) chosen based on validation performance, but more analysis on their selection or stability would be beneficial.

---

> ### Author Response · Authors · 2025-06-03
>
> We sincerely thank the reviewer for taking the time and effort to review our work. Below, we provide additional experimental results and clarifications to address your concerns.
>
> **1. Heuristics and Hyperparameters**
>
> We agree with your point and acknowledge that our ATF method involves some degree of hyperparameter tuning. While an additional efficient and principled approach to hyperparameter selection would certainly be desirable, we currently rely on validation-based tuning and leave the development of such strategies to future work. Nevertheless, our method already significantly narrows the feature space by filtering down to TF features—reducing, for instance, around a total of 40k active features in the Dev set to a compact set of about top-30 TF features—thereby greatly limiting the search space. We are actively working on identifying more specific and consistent steering patterns within this reduced set, as we agree this direction is both promising and valuable.
>
> Additionally, we used a fixed threshold of p = 0.9 throughout our experiments. as we believe the definitions of TF and HF features are most meaningful when the threshold parameter p is close to 1, aligning with their intended motivations. In response to your comment, we examined a range of p values and found that the overall patterns remain stable (i.e., high-performing languages consistently exhibit higher TF-feature ratios than low-performing languages) under small perturbations around p = 1. The corresponding statistics are provided in Table 1 (We have also updated the terminology to “High-Performing Languages (HPL)” and “Low-Performing Languages (LPL),” and added two additional models—Gemma2-2B-IT and Llama 3.1-8B-Instruct—with more detailed explanation provided below.)
>
> |TF-Feature Ratio (%)||Bn|Ar|Hi|Id|De|Es|Fr|Pt|Zh|\| En|HPL AVG|LPL AVG|
> |-|:-:|:-:|:-:|:-:|:-:|:-:|:-:|:-:|:-:|:-:|:-:|:-:|:-:|
> |*Llama3.1-8B-Instruct*|
> |* p=0.9||5.21|14.80|10.59|13.29|18.73|18.17|16.58|17.09|22.86|**\|** 24.87|18.69|10.97|
> |* p=0.8||6.14|15.38|11.27|14.02|19.46|18.91|18.12|17.54|23.11|**\|** 25.56|19.43|11.70|
> |* p=0.7||8.85|17.74|13.46|16.48|21.33|21.57|19.95|20.99|24.26|**\|** 27.50|21.62|14.13|
> |* p=0.6||13.32|23.36|19.29|22.06|25.44|25.67|23.79|25.01|27.15|**\|** 30.16|25.41|19.51|
> |* p=0.5||17.34|25.03|21.79|24.25|27.06|27.68|25.13|26.43|28.00|**\|** 31.05|26.86|22.10|
> |* p=0.4||26.68|39.47|34.28|38.04|39.60|39.72|37.82|38.94|37.80|**\|** 42.93|38.78|34.62|
> |*Gemma2-2B-IT*|
> |* p=0.9||0.43|0.78|0.66|1.81|5.16|4.04|3.51|3.46|3.13|**\|** 5.50|3.86|0.92|
> |* p=0.8||1.39|2.82|2.58|4.86|8.71|7.07|6.56|6.20|5.70|**\|** 9.29|6.85|2.91|
> |* p=0.7||3.56|4.97|5.55|8.07|12.12|10.25|10.06|9.42|9.37|**\|** 12.25|10.24|5.54|
> |* p=0.6||7.38|9.90|9.18|13.40|16.51|14.77|13.71|13.64|14.54|**\|** 16.39|14.63|9.96|
> |* p=0.5||14.60|16.96|15.76|20.85|22.78|21.37|19.87|20.07|21.07|**\|** 22.82|21.03|17.04|
> |* p=0.4||25.59|24.68|23.16|27.26|29.09|27.36|25.99|26.42|26.96|**\|** 28.88|27.16|25.17|
> ||
>
> *Table 1: Effect of the hyperparameter p on TF and HF features. Note that the definitions align with their intended motivation when p is close to 1.*
>
> (Continued)

---

> > ### Author Response · Authors · 2025-06-03
> >
> > (Continued)
> >
> > **2. Additional Experiments Including English Results.**
> >
> > Regarding your feedback, we have added two additional models—Gemma2-2B-IT and Llama 3.1-8B-IT (using SAEs from layer 20 with width 65k from the Gemma scope, and layer 23 with width 131k from the Llama scope)—to assess the generalizability of our approach. We have also included English results in the tables below. We’re excited to share these encouraging findings.
> > |Llama3.1-8B-IT||Bn|Ar|Hi|Id|De|Es|Fr|Pt|Zh|\| En|HPL AVG|LPL AVG|
> > |-|:-:|:-:|:-:|:-:|:-:|:-:|:-:|:-:|:-:|:-:|:-:|:-:|:-:|
> > |TF-Feature Ratio (%)||5.21|14.80|10.59|13.29|18.73|18.17|16.58|17.09|22.86|**\|** 24.87|18.69|10.97|
> > |HF-Feature Ratio (%)||0.10|0.56|0.35|1.44|1.29|0.67|0.31|0.66|1.06|**\|** 1.41|0.80|0.61|
> > |**Gemma2-2B-IT**|
> > |TF-Feature Ratio (%)||0.43|0.78|0.66|1.81|5.16|4.04|3.51|3.46|3.13|**\|** 5.50|3.86|0.92|
> > |HF-Feature Ratio (%)||0.01|0.17|0.22|1.04|1.09|0.67|1.22|0.98|1.02|**\|** 1.19|1.00|0.36|
> > ||
> >
> > *Table 2: HPL exhibit higher TF and HF feature ratio than LPL on both models.*
> >
> > | *Gemma2-2B-IT*||||||||||||||
> > |-|:-:|:-:|:-:|:-:|:-:|:-:|:-:|:-:|:-:|:-:|-|:-:|:-:|
> > |**ARC**||Bn|Ar|Hi|Id|De|Es|Fr|Pt|Zh|\| En|HPL AVG|LPL AVG|
> > |Zero-Shot Baseline||32.70|48.67|45.02|58.92|58.88|63.29|63.95|63.55|62.95|**\|** 71.93|62.52|46.33|
> > |ETI||34.76|48.07|46.65|59.61|59.06|64.15|62.92|62.92|61.75|**\|** 71.93|62.16|47.17|
> > |EHP||33.73|49.27|46.99|59.52|59.14|64.32|63.52|64.49|63.04|**\|** 71.93|62.90|47.38|
> > |EHP+ATF||**34.76**|**50.47**|**47.85**|**60.89**|**60.00**|**65.01**|**64.55**|**64.92**|**63.52**|**\|** **72.45**|**63.60**|**48.49**|
> > ||
> > |**MMLU-STEM**||Bn|Ar|Hi|Id|De|Es|Fr|Pt|Zh|\|En|HPL AVG|LPL AVG|
> > |Zero-Shot Baseline||30.36|34.35|35.68|38.65|39.51|41.16|40.38|39.44|41.24|**\|** 50.63|40.35|34.76|
> > |ETI||31.92|34.66|35.68|38.11|40.30|40.92|40.69|39.51|39.83|**\|** 50.63|40.25|35.09|
> > |EHP||32.16|35.52|35.76|40.14|40.30|41.86|41.39|40.53|40.62|**\|** 50.63|40.94|35.89|
> > |EHP+ATF||**32.86**|**36.15**|**36.46**|**40.30**|**40.85**|**42.10**|**41.63**|**41.55**|**40.69**|**\|** **51.17**|**41.36**|**36.44**|
> > ||
> >
> > *Table 3: Results on Gemma2-2B-IT.*
> >
> > | *Llama3.1-8B-Instruct*||||||||||||||
> > |-|:-:|:-:|:-:|:-:|:-:|:-:|:-:|:-:|:-:|:-:|:-:|:-:|:-:|
> > |**ARC**||Bn|Ar|Hi|Id|De|Es|Fr|Pt|Zh|\| En|HPL AVG|LPL AVG|
> > |Zero-Shot Baseline||13.65|41.03|36.43|51.03|53.99|58.15|58.03|56.60|58.40|**\|** 72.45|57.03|35.54|
> > |ETI||10.82|43.86|34.19|49.91|53.48|57.71|56.48|55.75|59.18|**\|** 72.45|56.52|34.70|
> > |EHP||22.15|46.18|40.12|53.09|55.11|59.86|58.20|58.92|58.49|**\|** 72.45|58.12|40.38|
> > |EHP+ATF||**31.24**|**47.73**|**41.58**|**53.77**|**56.48**|**60.46**|**58.63**|**59.61**|**59.52**|**\|** **73.91**|**58.94**|**43.58**|
> > ||
> > |**MMLU-STEM**||Bn|Ar|Hi|Id|De|Es|Fr|Pt|Zh|\|En|HPL AVG|LPL AVG|
> > |Zero-Shot Baseline||16.43|32.32|32.55|36.15|40.77|43.43|42.18|41.63|41.39|**\|** 48.83|41.88|29.36|
> > |ETI||9.00|31.22|31.38|35.21|40.61|41.55|42.10|41.24|41.94|**\|** 48.83|41.49|26.70|
> > |EHP||26.29|35.13|35.05|38.89|41.24|44.52|43.04|42.57|42.18|**\|** 48.83|42.71|33.84|
> > |EHP+ATF||**30.67**|**36.62**|**37.25**|**39.91**|**42.57**|**45.62**|**44.60**|**43.51**|**43.11**|**\|** **49.84**|**43.88**|**36.11**|
> > ||
> >
> > *Table 4: Results on Llama3.1-8B-Instruct.*
> >
> > The key takeaways are as follows:
> > * The main patterns from the original paper—such as the higher TF and HF feature ratios for high-performing languages (Table 2) and the effectiveness of EHP and ATF (Tables 3 and 4)—are consistent across the new models.
> > * English, as the pivot and highest-performing language, shows the highest TF-feature ratio and the second-highest HF-feature ratio among all languages examined across both models (Table 2). Moreover, English performance can be also improved through ATF, which we believe is an encouraging result (Tables 3 and 4).
> >
> > These results indicate that our key findings from the original paper generalize across different model sizes and architectures, thereby reinforcing our main contribution—namely, that the concepts of TF and HF features provide novel and fresh insights into the origins of language performance gaps in LLMs.
> >
> > ---
> > Thank you once again for your thoughtful feedback and for dedicating your time to reviewing our work. We hope these additions have addressed your concerns and strengthened the paper. If you find the revisions satisfactory, we would sincerely appreciate your reconsideration of the rating.

---

### Official Review · Reviewer_BE52 · 2025-05-21

**Rating:** 6
**Confidence:** 4
**Ethics Flag:** 1

**Summary:**

This work performs analysis of crosslingual transfer in Gemma2-9b models via Sparse Auto Encoders. It identifies several important features for the model which they refer to as "Task Instruction focus" and "Heading focus" and demonstrate that these features are activated unevenly across languages, which impacts the final performce in lower-resource languages. Based on these findings they propose 2 strategies to improve performance in lower-resource languages: usage of English headings in the prompt and amplification of Task instruction features. They demonstrate that both of these techniques allow to improve performance on lower-resource languages.

**Reasons To Accept:**

- this work performs interesting analysis of crosslingual knowledge transfer in Gemma 2-9B models with Sparse auto encoders
- it provides some insights on what are good "prompt formulation" would be to favour knowledge transfer, which is important for less resourced languages. these insights could easily be applied to other models (althought this work does not provide evidence that it would transfer beyond gemma-2-9b models)
 - Task-focus feature amplification is an interesting strategy, and could be used for other features identified by SAE in the future (Eg. authors mention the "answer position bias feature" that they leave for the future work)

**Reasons To Reject:**

(rather weaknesses than reasons to reject)
- the conclusions in this work might not hold beyond gemma 2 models, I believe it is very specific in the way that multilinguality is taken into account during model training. It should be easy to check an impact of English headings and English Task instructions in other models which was not done in this work
- some of the  languages that are reffered as low-resource in this work are not so low-resource , it would be good to exlicit the criteria for how authors define "low resource"
- one of the findings of this work is that it is better to have prompt headings (eg. "Question:", "Answer:", etc.) in English rather than in the target language. It seems quite unnatural to translate those headings (if I were to perform these experiments , I would keep them in English), but it is good to have explicit confirmation for this.

---

> ### Author Response · Authors · 2025-06-03
>
> We sincerely appreciate the reviewer’s time and effort in evaluating our work. Below, we present additional experimental results and clarifications in response to your comments. We have also rearranged the sections for improved clarity and flow.
>
> **1. Response to Comment 3**
>
> Yes, we agree that using English headings from the start can be natural. However, as you pointed out, offering an explanation for why English headings lead to better performance is valuable. Our findings show that LLMs exhibit multiple HF-features when processing prompts, suggesting that headings play a significant role in how prompts are interpreted—thus providing a plausible explanation for the observed performance differences.
>
> **2. Response to Comment 2**
>
> Thank you for pointing this out. We initially categorized the languages primarily based on their task performance, supplemented by some preliminary knowledge of their presence in the pretraining data. Given your concern—and the fact that the exact distribution of languages in the model’s pretraining and instruction tuning data is difficult to determine—we are happy to revise the terminology to 'high-performing' and 'low-performing' languages (HPL and LPL) with respect to a specific task.
>
> (Continued)

---

> > ### Author Response · Authors · 2025-06-03
> >
> > (Continued)
> >
> > **3. Response to Comment 1**
> >
> > Regarding your feedback, we have added two additional models—Gemma2-2B-IT and Llama 3.1-8B-IT (using SAEs from layer 20 with width 65k from the Gemma scope, and layer 23 with width 131k from the Llama scope)—to assess the generalizability of our approach. We have also included English results in the tables below. We’re excited to share these encouraging findings.
> > |Llama3.1-8B-IT||Bn|Ar|Hi|Id|De|Es|Fr|Pt|Zh|\| En|HPL AVG|LPL AVG|
> > |-|:-:|:-:|:-:|:-:|:-:|:-:|:-:|:-:|:-:|:-:|:-:|:-:|:-:|
> > |TF-Feature Ratio (%)||5.21|14.80|10.59|13.29|18.73|18.17|16.58|17.09|22.86|**\|** 24.87|18.69|10.97|
> > |HF-Feature Ratio (%)||0.10|0.56|0.35|1.44|1.29|0.67|0.31|0.66|1.06|**\|** 1.41|0.80|0.61|
> > |**Gemma2-2B-IT**|
> > |TF-Feature Ratio (%)||0.43|0.78|0.66|1.81|5.16|4.04|3.51|3.46|3.13|**\|** 5.50|3.86|0.92|
> > |HF-Feature Ratio (%)||0.01|0.17|0.22|1.04|1.09|0.67|1.22|0.98|1.02|**\|** 1.19|1.00|0.36|
> > ||
> >
> > *Table 1: HPL exhibit higher TF and HF feature ratio than LPL on both models.*
> >
> > | *Gemma2-2B-IT*||||||||||||||
> > |-|:-:|:-:|:-:|:-:|:-:|:-:|:-:|:-:|:-:|:-:|-|:-:|:-:|
> > |**ARC**||Bn|Ar|Hi|Id|De|Es|Fr|Pt|Zh|\| En|HPL AVG|LPL AVG|
> > |Zero-Shot Baseline||32.70|48.67|45.02|58.92|58.88|63.29|63.95|63.55|62.95|**\|** 71.93|62.52|46.33|
> > |ETI||34.76|48.07|46.65|59.61|59.06|64.15|62.92|62.92|61.75|**\|** 71.93|62.16|47.17|
> > |EHP||33.73|49.27|46.99|59.52|59.14|64.32|63.52|64.49|63.04|**\|** 71.93|62.90|47.38|
> > |EHP+ATF||**34.76**|**50.47**|**47.85**|**60.89**|**60.00**|**65.01**|**64.55**|**64.92**|**63.52**|**\|** **72.45**|**63.60**|**48.49**|
> > ||
> > |**MMLU-STEM**||Bn|Ar|Hi|Id|De|Es|Fr|Pt|Zh|\|En|HPL AVG|LPL AVG|
> > |Zero-Shot Baseline||30.36|34.35|35.68|38.65|39.51|41.16|40.38|39.44|41.24|**\|** 50.63|40.35|34.76|
> > |ETI||31.92|34.66|35.68|38.11|40.30|40.92|40.69|39.51|39.83|**\|** 50.63|40.25|35.09|
> > |EHP||32.16|35.52|35.76|40.14|40.30|41.86|41.39|40.53|40.62|**\|** 50.63|40.94|35.89|
> > |EHP+ATF||**32.86**|**36.15**|**36.46**|**40.30**|**40.85**|**42.10**|**41.63**|**41.55**|**40.69**|**\|** **51.17**|**41.36**|**36.44**|
> > ||
> >
> > *Table 2: Results on Gemma2-2B-IT.*
> >
> > | *Llama3.1-8B-Instruct*||||||||||||||
> > |-|:-:|:-:|:-:|:-:|:-:|:-:|:-:|:-:|:-:|:-:|:-:|:-:|:-:|
> > |**ARC**||Bn|Ar|Hi|Id|De|Es|Fr|Pt|Zh|\| En|HPL AVG|LPL AVG|
> > |Zero-Shot Baseline||13.65|41.03|36.43|51.03|53.99|58.15|58.03|56.60|58.40|**\|** 72.45|57.03|35.54|
> > |ETI||10.82|43.86|34.19|49.91|53.48|57.71|56.48|55.75|59.18|**\|** 72.45|56.52|34.70|
> > |EHP||22.15|46.18|40.12|53.09|55.11|59.86|58.20|58.92|58.49|**\|** 72.45|58.12|40.38|
> > |EHP+ATF||**31.24**|**47.73**|**41.58**|**53.77**|**56.48**|**60.46**|**58.63**|**59.61**|**59.52**|**\|** **73.91**|**58.94**|**43.58**|
> > ||
> > |**MMLU-STEM**||Bn|Ar|Hi|Id|De|Es|Fr|Pt|Zh|\|En|HPL AVG|LPL AVG|
> > |Zero-Shot Baseline||16.43|32.32|32.55|36.15|40.77|43.43|42.18|41.63|41.39|**\|** 48.83|41.88|29.36|
> > |ETI||9.00|31.22|31.38|35.21|40.61|41.55|42.10|41.24|41.94|**\|** 48.83|41.49|26.70|
> > |EHP||26.29|35.13|35.05|38.89|41.24|44.52|43.04|42.57|42.18|**\|** 48.83|42.71|33.84|
> > |EHP+ATF||**30.67**|**36.62**|**37.25**|**39.91**|**42.57**|**45.62**|**44.60**|**43.51**|**43.11**|**\|** **49.84**|**43.88**|**36.11**|
> > ||
> >
> > *Table 3: Results on Llama3.1-8B-Instruct.*
> >
> > The key takeaways are as follows:
> > * The main patterns from the original paper—such as the higher TF and HF feature ratios for high-performing languages (Table 1) and the effectiveness of EHP and ATF (Tables 2 and 3)—are consistent across the new models.
> > * English, as the pivot and highest-performing language, shows the highest TF-feature ratio and the second-highest HF-feature ratio among all languages examined across both models (Table 1). Moreover, English performance can be also improved through ATF, which we believe is an encouraging result (Tables 2 and 3).
> >
> > These results indicate that our key findings from the original paper generalize across different model sizes and architectures, thereby reinforcing our main contribution—namely, that the concepts of TF and HF features provide novel and fresh insights into the origins of language performance gaps in LLMs.
> >
> > ---
> > Once again, thank you for your thoughtful feedback and for taking the time to review our work. We hope these additions have addressed your concerns and improved the paper. If the revisions meet your expectations, we would deeply appreciate your reconsideration of the rating.

---

> > > ### Comment · Reviewer_BE52 · 2025-06-09
> > > **Thank you for your answer**
> > >
> > > Dear authors,
> > >
> > > thank you for your answers and additional experiments.
> > >
> > > It is good to see that ATF does improve  performance even in English.
> > >
> > > > Given your concern—and the fact that the exact distribution of languages in the model’s pretraining and instruction tuning data is difficult to determine—we are happy to revise the terminology to 'high-performing' and 'low-performing' languages (HPL and LPL) with respect to a specific task.
> > >
> > > I guess if you want to use this kind of terminology you still need to provide exact definition of what you consider as high-performing vs low-performing. I would assume that this would also be different for different models. Special case of Indonesian demonstrates that it  exhibits sometimes higher percentage of TF compared to "so-called HPL" languages.  Without clear definition the conclusions about HPL vs LPL sounds a bit weak (even though it makes perfect sense that lower percentage of TF features would lead to lower performance).
> > >
> > >
> > > Given the above concerns I do stick to my initial score. This work does have interesting findings but I feel like some rigour is missing in the experimental settings and the way the findings are presented.

---

> > > > ### Author Response · Authors · 2025-06-10
> > > >
> > > > Dear Reviewer,
> > > >
> > > > Thank you for your follow-up—we're glad to hear that you find our work interesting! We'd like to briefly add one more comment:
> > > >
> > > > Based on your feedback, we’ve carefully reconsidered how to improve our presentation. Since our goal is not to classify languages into groups, but rather to provide insights into the underlying differences in their performance, we believe it is more appropriate to reframe our presentation to emphasize that the **TF-feature ratio is strongly correlated with task performance.** Indeed, using the numbers reported in our initial response, we computed correlation coefficients of 0.928 for LLaMA3.1-8B-Instruct, 0.844 for Gemma2-2B-IT—both indicating a strong relationship. We will revise the draft accordingly and hope that this new framing presents our findings more clearly and with less ambiguity.
> > > >
> > > > ---
> > > > Thank you once again for your thoughtful feedback and engagement.
> > > >
> > > > Sincerely,
> > > >
> > > > The Authors

---

### Official Review · Reviewer_1FdK · 2025-05-23

**Rating:** 6
**Confidence:** 3
**Ethics Flag:** 1

**Summary:**

The paper applies sparse auto-encoders (SAEs) to interpret LLM embeddings in a cross-lingual
context. It introduces the concepts task instruction-focused and heading-focused SAE features and
applies them to interpret differences between high-resource and low-resource languages. The
method improves zero-shot performance in low-resource languages according to multilingual-MMLU.

**Reasons To Accept:**

The paper is very well written. The method Amplifying Task-Focused features improves
performance in both low-resource and high-resource languages. The paper helps to
explain the gap between low-resource and high-resource languages by showing that
LLMs have relatively fewer task-focused features for low-resource languages. Although
only one model is used, a total of 9 languages are examined.

**Reasons To Reject:**

The primary limitation is in the breadth of experiments: one LLM is tested, the
experiments are performed in a zero-shot context, and only one layer of the LLM is
examined.

---

> ### Author Response · Authors · 2025-06-03
>
> We sincerely thank the reviewer for taking the time and effort to evaluate our work. Below, we provide additional experimental results and clarifications in response to your comments.
>
> **Experiments on Additional Models**
>
> Regarding your feedback, we have added two additional models—Gemma2-2B-IT and Llama 3.1-8B-IT (using SAEs from layer 20 with width 65k from the Gemma scope, and layer 23 with width 131k from the Llama scope)—to assess the generalizability of our approach. We have also included English results in the tables below. We’re excited to share these encouraging findings. (Additionally, we have updated the terminology to “High-Performing Languages (HPL)” and “Low-Performing Languages (LPL)”.)
> |Llama3.1-8B-IT||Bn|Ar|Hi|Id|De|Es|Fr|Pt|Zh|\| En|HPL AVG|LPL AVG|
> |-|:-:|:-:|:-:|:-:|:-:|:-:|:-:|:-:|:-:|:-:|:-:|:-:|:-:|
> |TF-Feature Ratio (%)||5.21|14.80|10.59|13.29|18.73|18.17|16.58|17.09|22.86|**\|** 24.87|18.69|10.97|
> |HF-Feature Ratio (%)||0.10|0.56|0.35|1.44|1.29|0.67|0.31|0.66|1.06|**\|** 1.41|0.80|0.61|
> |**Gemma2-2B-IT**|
> |TF-Feature Ratio (%)||0.43|0.78|0.66|1.81|5.16|4.04|3.51|3.46|3.13|**\|** 5.50|3.86|0.92|
> |HF-Feature Ratio (%)||0.01|0.17|0.22|1.04|1.09|0.67|1.22|0.98|1.02|**\|** 1.19|1.00|0.36|
> ||
>
> *Table 1: HPL exhibit higher TF and HF feature ratio than LPL on both models.*
>
> | *Gemma2-2B-IT*||||||||||||||
> |-|:-:|:-:|:-:|:-:|:-:|:-:|:-:|:-:|:-:|:-:|-|:-:|:-:|
> |**ARC**||Bn|Ar|Hi|Id|De|Es|Fr|Pt|Zh|\| En|HPL AVG|LPL AVG|
> |Zero-Shot Baseline||32.70|48.67|45.02|58.92|58.88|63.29|63.95|63.55|62.95|**\|** 71.93|62.52|46.33|
> |ETI||34.76|48.07|46.65|59.61|59.06|64.15|62.92|62.92|61.75|**\|** 71.93|62.16|47.17|
> |EHP||33.73|49.27|46.99|59.52|59.14|64.32|63.52|64.49|63.04|**\|** 71.93|62.90|47.38|
> |EHP+ATF||**34.76**|**50.47**|**47.85**|**60.89**|**60.00**|**65.01**|**64.55**|**64.92**|**63.52**|**\|** **72.45**|**63.60**|**48.49**|
> ||
> |**MMLU-STEM**||Bn|Ar|Hi|Id|De|Es|Fr|Pt|Zh|\|En|HPL AVG|LPL AVG|
> |Zero-Shot Baseline||30.36|34.35|35.68|38.65|39.51|41.16|40.38|39.44|41.24|**\|** 50.63|40.35|34.76|
> |ETI||31.92|34.66|35.68|38.11|40.30|40.92|40.69|39.51|39.83|**\|** 50.63|40.25|35.09|
> |EHP||32.16|35.52|35.76|40.14|40.30|41.86|41.39|40.53|40.62|**\|** 50.63|40.94|35.89|
> |EHP+ATF||**32.86**|**36.15**|**36.46**|**40.30**|**40.85**|**42.10**|**41.63**|**41.55**|**40.69**|**\|** **51.17**|**41.36**|**36.44**|
> ||
>
> *Table 2: Results on Gemma2-2B-IT.*
>
> | *Llama3.1-8B-Instruct*||||||||||||||
> |-|:-:|:-:|:-:|:-:|:-:|:-:|:-:|:-:|:-:|:-:|:-:|:-:|:-:|
> |**ARC**||Bn|Ar|Hi|Id|De|Es|Fr|Pt|Zh|\| En|HPL AVG|LPL AVG|
> |Zero-Shot Baseline||13.65|41.03|36.43|51.03|53.99|58.15|58.03|56.60|58.40|**\|** 72.45|57.03|35.54|
> |ETI||10.82|43.86|34.19|49.91|53.48|57.71|56.48|55.75|59.18|**\|** 72.45|56.52|34.70|
> |EHP||22.15|46.18|40.12|53.09|55.11|59.86|58.20|58.92|58.49|**\|** 72.45|58.12|40.38|
> |EHP+ATF||**31.24**|**47.73**|**41.58**|**53.77**|**56.48**|**60.46**|**58.63**|**59.61**|**59.52**|**\|** **73.91**|**58.94**|**43.58**|
> ||
> |**MMLU-STEM**||Bn|Ar|Hi|Id|De|Es|Fr|Pt|Zh|\|En|HPL AVG|LPL AVG|
> |Zero-Shot Baseline||16.43|32.32|32.55|36.15|40.77|43.43|42.18|41.63|41.39|**\|** 48.83|41.88|29.36|
> |ETI||9.00|31.22|31.38|35.21|40.61|41.55|42.10|41.24|41.94|**\|** 48.83|41.49|26.70|
> |EHP||26.29|35.13|35.05|38.89|41.24|44.52|43.04|42.57|42.18|**\|** 48.83|42.71|33.84|
> |EHP+ATF||**30.67**|**36.62**|**37.25**|**39.91**|**42.57**|**45.62**|**44.60**|**43.51**|**43.11**|**\|** **49.84**|**43.88**|**36.11**|
> ||
>
> *Table 3: Results on Llama3.1-8B-Instruct.*
>
> The key takeaways are as follows:
> * The main patterns from the original paper—such as the higher TF and HF feature ratios for high-performing languages (Table 1) and the effectiveness of EHP and ATF (Tables 2 and 3)—are consistent across the new models.
> * English, as the pivot and highest-performing language, shows the highest TF-feature ratio and the second-highest HF-feature ratio among all languages examined across both models (Table 1). Moreover, English performance can be also improved through ATF, which we believe is an encouraging result (Tables 2 and 3).
>
> These results indicate that our key findings from the original paper generalize across different model sizes and architectures, thereby reinforcing our main contribution—namely, that the concepts of TF and HF features provide novel and fresh insights into the origins of language performance gaps in LLMs.
>
> We focused on the zero-shot setting, as we believe it is better suited to revealing intrinsic differences in LLM behavior across languages, as discussed in lines 105–109 of the paper. We agree that analyzing additional layers could offer further insights, and we will include more layer-wise results during the remaining week of the rebuttal period.
>
> ---
> Thank you once again for your valuable feedback and for taking the time to review our work. We hope these additions have addressed your concerns and strengthened the paper. If you find the revisions satisfactory, we would deeply appreciate your reconsideration of the rating.

---

### Official Review · Reviewer_UmKf · 2025-05-26

**Rating:** 7
**Confidence:** 4
**Ethics Flag:** 1

**Summary:**

This paper explores how LLMs handle multilingual input, using sparse autoencoders as a tool for interpretability. The authors introduce two new feature types—task instruction–focused (TF) features and heading-focused (HF) features—that emerge in the sparse representations of LLM embeddings. These features reveal how models parse prompt structure, especially across languages with varying resource levels.
A key insight is that low-resource languages tend to lack TF-features, which correlates with their lower downstream performance. To address this, the authors propose two lightweight interventions:

(i) English Heading Prompts (EHP) – swapping out native-language headings like “Question” or “Choices” with their English counterparts,
(ii) Amplifying TF-Features (ATF) – boosting the influence of known TF-features in the model’s hidden states.

Together, these methods improve overall zero-shot performance and specifically by up to 3.7 points on multilingual MMLU and 1.85 on ARC in low-resource languages scenarios.

While the results are promising, some methodological choices raise open questions that merit more discussion.
First, the TF-features are derived only from the ARC training set but are also used to steer model behavior on m-MMLU, a dataset with notably different prompt structure and domain content. This constitutes an implicit form of cross-task feature transfer, which the paper doesn’t acknowledge or validate. Without confirming that the same features remain meaningful in m-MMLU, the effectiveness of ATF across tasks rests on a potentially fragile assumption.

Second, unless I’ve missed something, ATF is never evaluated in isolation. The reported gains always occur alongside EHP, making it hard to assess how much amplification alone contributes. Since the paper positions TF-features as central to performance gaps, a standalone ATF analysis would help clarify their actual impact.

Finally, the paper assumes structural consistency in how LLMs parse prompts across languages, but doesn’t explore how sensitive these features are to prompt phrasing or localization style. Given the variability in multilingual prompt design, this could be an important point.

These remarks don’t undermine the core contributions, but they do suggest that more careful handling of feature generalization and ablation completeness would strengthen this paper.


I found this paper well written, but a bit dense at times, so to conclude, I suggest this paper to be accepted.

Notes:
* some of the parameters choice seem a bit arbitrary (eg. see section 4.5, why |DTF| was set to 3, why the alpha was set between 50 and 500 ?
* a contemporary paper just came out with a similar use of SAEs, maybe in a final version, you could discuss it?
https://arxiv.org/pdf/2505.05111
(Deng et al, 2025/05/08)

**Questions To Authors:**

the questions are pretty much self-explanotory given the above but here we go:
- Why didn't you perform an experiment to assess the impact of ATF alone.
- what can you tell on the fact that you extracted the TF-features from ARC and applied to another data set?

**Reasons To Accept:**

- Well written paper, original use of SAEs for multilingual analysis
- Good experiments design that lead to convincing results
- Practical interventions with measurable gains

**Reasons To Reject:**

None that I can see, I would have just liked to see:
- more elements regarding the impact of the ATF technique
- a discussion on the sensitivity of the features to prompt variability (monolingual and multilingual)
- a discussion on the cross-task generalization you've implicitely done with the TF-Features extracted from ARC and applied to m-mllu  (the last two questions are ofc linked)

---

> ### Author Response · Authors · 2025-06-03
>
> We sincerely thank the reviewer for taking the time and effort to evaluate our work. We are delighted to hear that you found our work interesting! We summarized how we have addressed your comments below.
>
> **1. About the Reference**
>
> Thank you for sharing the reference. We reviewed it and found it quite interesting. It shares similarities with our work in that it also explores multilingualism in LLMs through the lens of SAEs. Their approach identifies features with higher average activation for specific languages and uses them for ablation, demonstrating that removing these features primarily degrades performance (i.e., increases CE loss) for the corresponding language while leaving others relatively unaffected. They also show that these language-specific features can be used to steer outputs toward a particular language, which is a direct application of their findings.
>
> We believe the main difference of our work is its **more analytical focus**—aimed at gaining a deeper understanding of how LLMs process the prompts and the origins of performance gaps between languages. Our key contribution lies in introducing the new concepts of TF and HF features, which exhibit consistent distributional differences that help explain disparities in language performance.
>
> **2. About Hyperparameters**
>
> We agree with your point and acknowledge that our ATF method involves some degree of hyperparameter tuning. In practice, we found that the presented hyperparameters generally work well across different languages and tasks. While developing a more efficient and principled approach to hyperparameter selection would certainly be beneficial, we currently rely on validation-based tuning and leave more robust strategies for future work. Nevertheless, we would like to emphasize that our method already substantially reduces the feature space by narrowing it down to just a few dozen TF features, significantly limiting the search space. We are actively working to identify more specific and consistent steering patterns within this reduced set, with the goal of further minimizing reliance on hyperparameter tuning.
>
> **3. ATF-only performance**
>
> Thank you for pointing this out. Our primary focus was to assess whether ATF could provide additional gains on top of the already strong EHP prompt. However, we agree that assessing the standalone performance of ATF is also important. Due to the heavy load of additional experiments (across six reviewers), we currently report initial ATF-only results on the Llama3.1-8B-Instruct model for the ARC task (N.B., based on suggestions from other reviewers, we have added two more models—Gemma2-2B-IT and Llama3.1-8B-Instruct). We observe in Table 1 that while ATF yields a notable improvement over the zero-shot baseline, it still underperforms compared to EHP and EHP+ATF. We plan to provide more results over the remainder of the rebuttal period.
>
> | *Llama3.1-8B-Instruct*||||||||||||||
> |-|:-:|:-:|:-:|:-:|:-:|:-:|:-:|:-:|:-:|:-:|:-:|:-:|:-:|
> |**ARC**||Bn|Ar|Hi|Id|De|Es|Fr|Pt|Zh|\| En|HPL AVG|LPL AVG|
> |Zero-Shot Baseline||13.65|41.03|36.43|51.03|53.99|58.15|58.03|56.60|58.40|**\|** 72.45|57.03|35.54|
> |ATF||23.35|44.89|39.52|51.46|54.33|59.01|58.20|58.06|**59.86**|**\|** 73.91|57.89|39.81|
> |ETI||10.82|43.86|34.19|49.91|53.48|57.71|56.48|55.75|59.18|**\|** 72.45|56.52|34.70|
> |EHP||22.15|46.18|40.12|53.09|55.11|59.86|58.20|58.92|58.49|**\|** 72.45|58.12|40.38|
> |EHP+ATF||**31.24**|**47.73**|**41.58**|**53.77**|**56.48**|**60.46**|**58.63**|**59.61**|59.52|**\|** **73.91**|**58.94**|**43.58**|
> ||
>
> *Table 1. ATF-only results on Llama3.1-8B-Instruct for ARC (row 2).*
>
> **4. TF-Feature Transfer and Prompt Sensitivity**
>
> We appreciate this insightful observation. First, we would like to clarify that we used the same task instruction and prompt structure for both MMLU and ARC—for example, the English instruction: “Answer the following multiple-choice question, ensuring the response is one of A, B, C, or D.” As a result, the TF features are largely overlapping across the two tasks. For instance, using Gemma2-2B-IT, we identified 39 TF-features for ARC and 34 for MMLU, with 32 shared features. Consequently, the ATF results are essentially very similar. In summary, because we used the same task instruction and prompt structure with a high threshold of p = 0.9, the features captured from each dataset show a high degree of overlap. We agree, however, that using different or more task-specific instructions could lead to variations in TF features, and exploring this would be a valuable direction for future work. Additionally, investigating the robustness of TF features under paraphrased instructions is another promising avenue that we plan to pursue.
>
> ---
> We hope these additions have addressed your concerns. Thank you again for your intriguing feedback and for taking the time to review our work!

---

> > ### Comment · Reviewer_UmKf · 2025-06-08
> >
> > Thanks for your answers and the new experiments. Frankly, I was not aware that providing a full range of new experiments was allowed. I thought that only answers and clarifications when needed was authorized.
> >
> > Not sure if you'll have the time to answer but do you have an intuition as to why the ATF feature seems to work that better than EHP on Chinese and English ? if you were to rank your languages in terms of morphological-richness (Tsarfaty et al, 2010, SPMRL) you could see a pattern here.

---

> > > ### Author Response · Authors · 2025-06-10
> > >
> > > Dear Reviewer,
> > >
> > > Thank you for your follow-up.
> > > Regarding your question, we would first like to clarify that EHP is identical to the baseline in the English setting, since English already uses English headings. In the case of Chinese, the results show that the baseline and EHP perform similarly, and likewise, Baseline+ATF and EHP+ATF yield comparable performance. We believe this suggests that EHP simply does not contribute much for Chinese in LLaMA3.1-8B-Instruct. As for why EHP is not effective for Chinese, we speculate that the model has already learned to effectively utilize the functionality of headings in this language, which may reduce the benefit of EHP. This aligns with our broader observation that EHP tends to be slightly more beneficial for low-performing languages than for high-performing ones.
> > >
> > > Thank you for sharing the reference. While we don’t yet have a clear intuition about how EHP might relate to morphological richness, we will reflect on this further and aim to incorporate any relevant insights (e.g., tokenization or SAE-feature overlap between English and Chinese headings) into the revised version.
> > >
> > > Thank you once again for your thoughtful feedback and engagement.
> > >
> > > Sincerely,
> > >
> > > The Authors

---

### Official Review · Reviewer_mB9c · 2025-05-27

**Rating:** 6
**Confidence:** 3
**Ethics Flag:** 1

**Summary:**

In this paper the authors explore multilingual capabilities of LLMs using sparse autoencoders (SAE). They introduce task instruction-focused (TF) and heading-focused (HF) SAE features and leverage these to study differences in zero-shot performance across 9 different languages. Building on these two features they introduce to techniques to potentially increase performance by incorporating English keywords in headings and by amplifying the introduced TF-features.

**Questions To Authors:**

1. Please find another notation for the activation of feature $t$, I don't know if this was on purpose, but $S_E(X_i)_t$ is not appropriate as a notation
2. While not currently published, there has been some recent criticism on SAE: https://deepmindsafetyresearch.medium.com/negative-results-for-sparse-autoencoders-on-downstream-tasks-and-deprioritising-sae-research-6cadcfc125b9. Could you mention how this impacts the results of your work?
3. Please mention the languages that you're working on from the beginning of the paper, not in page 5. And also avoid calling Arabic, Hindi, Bengali and Indonesian low-resource languages. This is not true. Something like low-performance languages (at least for your selected tasks) would be more appropriate
4. Could you expand the discussion in the limitation section about your study only working with Gemma2? How do you think your study translates to other models?

**Reasons To Accept:**

1. The paper is clear and well-written
2. The authors use SAE to introduce two concepts and then use these to sucessfully increase performance on zero-shot ARC and MMLU
3. The method seems to work across multiple languages
4.  The introduced TF and HF-features seem to have interpretability value according to their experiments

**Reasons To Reject:**

1. The paper claims to improve performance on low resource languages, while not actually dealing with any low resource language. Arabic has almost 500 million speakers worldwide, Hindi has more than 600 million, Indonesian 252 million and Bengali around 284 million. For comparison, one of the high resource languages used by the authors, German, has around 180 million speakers
2. The two benchmarks used are translated for English, which could explain why the English Heading approach worked so well, however this is never mentioned or addressed by the authors
3. The authors also failed to mention recent criticism on SAE approaches and how this affects their work
4. The study by design is only limited to one model, which makes the conclusions of the study difficult to translate to other models

---

> ### Author Response · Authors · 2025-06-03
>
> We thank the reviewer for the valuable feedback. We would like to provide additional experimental results and explanations to address your concerns. We refer to items under “Questions to Authors” as Q and those under “Reasons to Reject” as R.
>
> **1. Response to Q#1.**
>
> Thank you for the suggestion. We will replace the notation with clearer ones such as $S_E^{(t)}(X_i)$.
>
> **2. Response to Q#2 and R#3.**
>
> Yes, we are aware that the practical utility of SAEs has received mixed reviews in recent literature. However, most of the criticism has centered on their use in probing tasks—as discussed in the link you provided and in https://arxiv.org/pdf/2502.16681 or https://arxiv.org/abs/2501.17148. While we agree that replacing hidden states with SAE activations may not offer performance gains, we believe that SAEs can add value when used as **analytical tools and assistants** rather than as direct replacements.
>
> For example, our introduction of TF and HF features demonstrates how SAEs can reveal new, fine-grained insights into how LLMs process prompts—insights that would be difficult to uncover without them. This analytical strength is where we believe SAEs truly shine. While the use of SAEs for model steering has also received mixed opinions, the overall sentiment appears to lean slightly optimistic, and we believe our work contributes positively to this growing perspective.
>
> **3. Response to Q#3 and R#1.**
>
> Thank you for pointing this out. We are happy to revise the terminology to "high-performing" and "low-performing" languages (HPL and LPL) with respect to a specific task. Since the precise distribution of languages in the model’s pretraining and instruction tuning data is difficult to determine, we agree that classifying languages based on their performance on a given task is the most straightforward and simple approach.
>
> **4. Response to R#2.**
>
> We’re not entirely sure why using data translated from English would necessarily make EHP more effective. EHP simply modifies three heading tokens—“Question,” “Choices,” and “Answer”—into English, which are fairly common words and not specifically tied to the original English phrasing of the task. We would greatly appreciate further clarification if we’re missing a particular concern or nuance here!
>
> (Continued)

---

> > ### Author Response · Authors · 2025-06-03
> >
> > **5. Response to Q#4 and R#4.**
> >
> > Regarding your feedback, we have added two additional models—Gemma2-2B-IT and Llama 3.1-8B-IT (using SAEs from layer 20 with width 65k from the Gemma scope, and layer 23 with width 131k from the Llama scope)—to assess the generalizability of our approach. We have also included English results in the tables below. We are excited to share these encouraging findings.
> > |Llama3.1-8B-IT||Bn|Ar|Hi|Id|De|Es|Fr|Pt|Zh|\| En|HPL AVG|LPL AVG|
> > |-|:-:|:-:|:-:|:-:|:-:|:-:|:-:|:-:|:-:|:-:|:-:|:-:|:-:|
> > |TF-Feature Ratio (%)||5.21|14.80|10.59|13.29|18.73|18.17|16.58|17.09|22.86|**\|** 24.87|18.69|10.97|
> > |HF-Feature Ratio (%)||0.10|0.56|0.35|1.44|1.29|0.67|0.31|0.66|1.06|**\|** 1.41|0.80|0.61|
> > |**Gemma2-2B-IT**|
> > |TF-Feature Ratio (%)||0.43|0.78|0.66|1.81|5.16|4.04|3.51|3.46|3.13|**\|** 5.50|3.86|0.92|
> > |HF-Feature Ratio (%)||0.01|0.17|0.22|1.04|1.09|0.67|1.22|0.98|1.02|**\|** 1.19|1.00|0.36|
> > ||
> >
> > *Table 1: HPL exhibit higher TF and HF feature ratio than LPL on both models.*
> >
> > | *Gemma2-2B-IT*||||||||||||||
> > |-|:-:|:-:|:-:|:-:|:-:|:-:|:-:|:-:|:-:|:-:|-|:-:|:-:|
> > |**ARC**||Bn|Ar|Hi|Id|De|Es|Fr|Pt|Zh|\| En|HPL AVG|LPL AVG|
> > |Zero-Shot Baseline||32.70|48.67|45.02|58.92|58.88|63.29|63.95|63.55|62.95|**\|** 71.93|62.52|46.33|
> > |ETI||34.76|48.07|46.65|59.61|59.06|64.15|62.92|62.92|61.75|**\|** 71.93|62.16|47.17|
> > |EHP||33.73|49.27|46.99|59.52|59.14|64.32|63.52|64.49|63.04|**\|** 71.93|62.90|47.38|
> > |EHP+ATF||**34.76**|**50.47**|**47.85**|**60.89**|**60.00**|**65.01**|**64.55**|**64.92**|**63.52**|**\|** **72.45**|**63.60**|**48.49**|
> > ||
> > |**MMLU-STEM**||Bn|Ar|Hi|Id|De|Es|Fr|Pt|Zh|\|En|HPL AVG|LPL AVG|
> > |Zero-Shot Baseline||30.36|34.35|35.68|38.65|39.51|41.16|40.38|39.44|41.24|**\|** 50.63|40.35|34.76|
> > |ETI||31.92|34.66|35.68|38.11|40.30|40.92|40.69|39.51|39.83|**\|** 50.63|40.25|35.09|
> > |EHP||32.16|35.52|35.76|40.14|40.30|41.86|41.39|40.53|40.62|**\|** 50.63|40.94|35.89|
> > |EHP+ATF||**32.86**|**36.15**|**36.46**|**40.30**|**40.85**|**42.10**|**41.63**|**41.55**|**40.69**|**\|** **51.17**|**41.36**|**36.44**|
> > ||
> >
> > *Table 2: Results on Gemma2-2B-IT.*
> >
> > | *Llama3.1-8B-Instruct*||||||||||||||
> > |-|:-:|:-:|:-:|:-:|:-:|:-:|:-:|:-:|:-:|:-:|:-:|:-:|:-:|
> > |**ARC**||Bn|Ar|Hi|Id|De|Es|Fr|Pt|Zh|\| En|HPL AVG|LPL AVG|
> > |Zero-Shot Baseline||13.65|41.03|36.43|51.03|53.99|58.15|58.03|56.60|58.40|**\|** 72.45|57.03|35.54|
> > |ETI||10.82|43.86|34.19|49.91|53.48|57.71|56.48|55.75|59.18|**\|** 72.45|56.52|34.70|
> > |EHP||22.15|46.18|40.12|53.09|55.11|59.86|58.20|58.92|58.49|**\|** 72.45|58.12|40.38|
> > |EHP+ATF||**31.24**|**47.73**|**41.58**|**53.77**|**56.48**|**60.46**|**58.63**|**59.61**|**59.52**|**\|** **73.91**|**58.94**|**43.58**|
> > ||
> > |**MMLU-STEM**||Bn|Ar|Hi|Id|De|Es|Fr|Pt|Zh|\|En|HPL AVG|LPL AVG|
> > |Zero-Shot Baseline||16.43|32.32|32.55|36.15|40.77|43.43|42.18|41.63|41.39|**\|** 48.83|41.88|29.36|
> > |ETI||9.00|31.22|31.38|35.21|40.61|41.55|42.10|41.24|41.94|**\|** 48.83|41.49|26.70|
> > |EHP||26.29|35.13|35.05|38.89|41.24|44.52|43.04|42.57|42.18|**\|** 48.83|42.71|33.84|
> > |EHP+ATF||**30.67**|**36.62**|**37.25**|**39.91**|**42.57**|**45.62**|**44.60**|**43.51**|**43.11**|**\|** **49.84**|**43.88**|**36.11**|
> > ||
> >
> > *Table 3: Results on Llama3.1-8B-Instruct.*
> >
> > The key takeaways are as follows:
> > * The main patterns from the original paper—such as the higher TF and HF feature ratios for high-performing languages (Table 1) and the effectiveness of EHP and ATF (Tables 2 and 3)—are consistent across the new models.
> > * English, as the pivot and highest-performing language, shows the highest TF-feature ratio and the second-highest HF-feature ratio among all languages examined across both models (Table 1). Moreover, English performance can be also improved through ATF, which we believe is an encouraging result (Tables 2 and 3).
> >
> > These results indicate that our key findings from the original paper generalize across different model sizes and architectures, thereby reinforcing our main contribution—namely, that the concepts of TF and HF features provide novel and fresh insights into the origins of language performance gaps in LLMs.
> >
> > ---
> > Thank you once again for taking the time and effort to review our work. We hope these additions have addressed your concerns and strengthened the paper. If you find the revisions satisfactory, we would deeply appreciate your reconsideration of the rating.

---

> ### Comment · Reviewer_mB9c · 2025-06-07
>
> I thank the author for the response.
>
> I have updated the score to better reflect my opinion after your comment. The new experiments are useful, however I still stand by the criticism of SAEs and I think this is something that should be better discussed in the main body of the paper.
>
> Regarding the notation, it was already clear. Just refrain from using notation that deliberately spell things like "SEX" or "SEXi".

---

> > ### Author Response · Authors · 2025-06-10
> >
> > Dear Reviewer,
> >
> > Thank you for your follow-up! We agree that the results regarding the effectiveness of SAEs have been mixed. In our revised paper, we will more clearly explain how our approach differs from prior unsuccessful use cases. Thank you again for your valuable engagement in the review process.
> >
> > Sincerely,
> >
> > The Authors

---

### Decision · Program_Chairs · 2025-07-08

**Decision:**

Accept

**Comment:**

This paper investigates how large language models (LLMs) handle multilingual prompts by leveraging sparse autoencoders (SAEs) to interpret model behavior. The authors introduce two new feature types: task instruction–focused (TF) and heading-focused (HF) features, which help explain performance differences across languages, especially between high- and low-resource ones. They propose simple methods like using English headings and boosting TF features, which improve performance on zero-shot tasks like ARC and m-MMLU across nine languages. While the study focuses on one model (Gemma 2.9B), the findings offer useful insights into prompt design and could be applied more broadly in future work.

Overall, the reviewers agree that the paper is well-written and presents practical applications across multiple languages, achieving performance improvements of up to 3.7% in low-resource settings. The analysis is clearly presented and demonstrates that sparse autoencoders (SAEs) can offer valuable insights into language prompt structure.

Given the quality of the work and its potential impact, I recommend the paper for acceptance. I encourage the authors to thoughtfully address the reviewers’ comments to further enhance the strength and clarity of the paper.

Possible improvements:
- Improve the clarity of the figures (the texts on Figures 1 and 2 are too hard to read).
- Fix the notation.